# The timing of unprecedented hydrological drought under climate change

Yusuke Satoh [1,2,3✉], Kei Yoshimura [4], Yadu Pokhrel [5], Hyungjun Kim [2,4,6], Hideo Shiogama [1], Tokuta Yokohata [1], Naota Hanasaki [1], Yoshihide Wada [3,7], Peter Burek [3], Edward Byers [3], Hannes Müller Schmied [8,9], Dieter Gerten [10,11], Sebastian Ostberg [10], Simon Newland Gosling [12], Julien Eric Stanslas Boulange[1] & Taikan Oki [13]

Droughts that exceed the magnitudes of historical variation ranges could occur increasingly frequently under future climate conditions. However, the time of the emergence of unprecedented drought conditions under climate change has rarely been examined. Here, using multimodel hydrological simulations, we investigate the changes in the frequency of hydrological drought (defined as abnormally low river discharge) under high and low greenhouse gas concentration scenarios and existing water resource management measures and estimate the time of the first emergence of unprecedented regional drought conditions centered on the low-flow season. The times are detected for several subcontinental-scale regions, and three regions, namely, Southwestern South America, Mediterranean Europe, and Northern Africa, exhibit particularly robust results under the high-emission scenario. These three regions are expected to confront unprecedented conditions within the next 30 years with a high likelihood regardless of the emission scenarios. In addition, the results obtained herein demonstrate the benefits of the lower-emission pathway in reducing the likelihood of emergence. The Paris Agreement goals are shown to be effective in reducing the likelihood to the unlikely level in most regions. However, appropriate and prior adaptation measures are considered indispensable when facing unprecedented drought conditions. The results of this study underscore the importance of improving drought preparedness within the considered time horizons.

[1] National Institute for Environmental Studies, Tsukuba, Japan. [2] Moon Soul Graduate School of Future Strategy, Korea Advanced Institute of Science and Technology, Daejeon, Korea. [3] International Institute for Applied Systems Analysis, Laxenburg, Austria. [4] Institute of Industrial Science, The University of Tokyo, Tokyo, Japan. [5] Department of Civil and Environmental Engineering, Michigan State University, Michigan, USA. [6] Department of Civil and Environmental Engineering, Korea Advanced Institute of Science and Technology, Daejeon, Korea. [7] Department of Physical Geography, Utrecht University, Utrecht, Netherlands. [8] Institute of Physical Geography, Goethe-University Frankfurt, Frankfurt am Main, Germany. [9] Senckenberg Leibniz Biodiversity and Climate Research Centre Frankfurt, Frankfurt am Main, Germany. [10] Potsdam Institute for Climate Impact Research, Member of the Leibniz Association, Potsdam, Germany. [11] Geography Department, Humboldt-Universität zu Berlin, Berlin, Germany. [12] School of Geography, University of Nottingham, Nottingham, UK. [13] Graduate School of Engineering, The University of Tokyo, Tokyo, Japan. ✉email: yusuke.satoh@kaist.ac.kr

The intensification of the hydrological cycle under climate change is projected to exacerbate future drought conditions in many parts of the world due to changes in precipitation patterns, altered snow accumulation and melt regimes, and increases in evapotranspiration[1–7]. An improved understanding of the emergence of more severe future drought conditions is imperative to cope with potential increases in drought exposure. It is thus crucial to understand how unprecedented drought—drought conditions that exceed the magnitude of past drought occurrences—will evolve, as the hydrological stationarity assumption adopted in past decades is expected to be inappropriate for addressing future water management[8]. The existing hydrological, agricultural, and industrial infrastructures designed for water resource management and the management strategies have been planned based on historical experience and statistics. Obtaining an understanding of the time at which drought conditions cross into hitherto unprecedented stages beyond stationary conditions is thus indispensable for ensuring effective adaptation plans and mitigation strategies. With similar motivations and/or for detecting climate change, the concept of the time of emergence (ToE) of climate change has been developed. In general, ToE is defined as the time at which a climate change signal emerges from natural variability, indicating the beginning of a new regime. Previous studies have applied the ToE framework to study hydroclimatic variables[9–11], fire[12], and biodiversity[13], as well as average[14–21] or extreme[22–25] conditions of temperature or precipitation; several different approaches have been proposed to detect the relevant significant shifts.

However, the ToE of unprecedented drought conditions under the background of a warmer climate has rarely been examined. Future drought conditions expected under climate change have been extensively studied in general, but most large-scale assessments of future drought conditions have applied time-slice[26–30] or temperature-slice[31–34] approaches to investigate drought changes for specific future periods or under certain global mean temperature rise magnitudes. Only a few studies[35–37] have evaluated the ToE of drought or low-flow conditions, but to the best of our knowledge, only the study by Touma et al.[6] has estimated projected ToE values for drought at a global scale; in their study, the authors compared the impacts of warming with the internal variability in the recent past. They employed future projections derived from general circulation models (GCMs) without bias correction for a single high greenhouse gas (GHG) emission scenario, Representative Concentration Pathway 8.5 (RCP8.5)[38], and discussed the ToEs concerning the spatial extents of four drought types in 26 subcontinental regions. However, despite projected significant changes, their study did not detect the emergence of robust changes regarding runoff drought and precipitation drought across any regions, although signals were detected for two evaporation-related drought indices.

Therefore, there is a need for drought ToE studies involving further considerations. First and most importantly, examining future drought conditions while covering a range of climate scenarios, including a low-emission scenario (RCP2.6), is critical in view of the 1.5 °C and well-below 2 °C targets of the Paris Agreement[39]. As RCP2.6 is a relatively low emission scenario among the established RCPs and its associated warming rate aligns with that considered in the Paris Agreement, this low-emission scenario enables near-term assessments while considering inevitable risk and the benefits of mitigation efforts. Since the literature indicates regionally significant drought intensification even at the 1.5 °C warming level[31–34], it is important to estimate the critical timing of drought under a low-emission scenario. In fact, almost all ToE studies have focused on only the high-emission scenario to obtain robust ToEs of climate change, and these studies constrained the definition of ToE to denote permanent exceedance; i.e., under this assumption, the change magnitude will not fall within the reference range of natural variability once the signal emerges. Touma et al. 2015 is not an exception to this approach. However, this strict constraint on the ToE definition may result in critical timing information being missed when estimating the urgency of adaptation and mitigation measures. Second, considering the propagation of water deficits from upstream to downstream regions is critical when performing drought assessments in view of water resource assessments[40]. Hence, it is essential to consider river discharge, especially abnormal low-flow conditions, when investigating future drought with regard to water, food, and energy security; in contrast, Touma et al.[6] focused on local runoff when examining future hydrological drought.

Here, we present the first estimation of the time of the first emergence (TFE) of unprecedented regional drought conditions that last over several consecutive years, thus providing a new quantification of urgency that can be applied to adaptation and mitigation strategies with regard to drought under climate change (Supplementary Fig. 1; see Methods). Notably, in contrast to the ToE, the TFE allows for recovery following years of consecutive unprecedented conditions and focuses on the information of the first emergence of unprecedented drought conditions. By exploring the annual evolution of the regional average drought frequency, the onset of unprecedented regional drought conditions is defined herein as a departure in which the time series of the regional average drought frequency exceeds the upper bounds of its historical climate variability consecutively for a certain number of years. This threshold is derived from the maximum value measured during the historical reference period (1865–2005). Our study is centered on the TFE of an unprecedented period equal to or more than five years in length (TFE$_5$); this process differs from the methods used in most preceding ToE studies that investigated permanent exceedance. Instead, we evaluate the uncertainty in TFE$_5$ that arises from irreducible natural climate variability as well as from the utilized model structures and provide the likelihood of TFE$_5$ over time. The uncertainty in TFE$_5$ is quantified based on 2 million time series resampled by the block-wise bootstrap method[41] (see Methods).

## Results

**Multi-climate and multi-impact model drought projection**. We investigate hydrological drought globally at a 0.5°×0.5° spatial resolution under a historical scenario (1861–2005) and under the RCP2.6 and RCP8.5 (2006–2099) future scenarios. Daily river discharge simulations obtained from five global water models (GWMs) forced by bias-corrected climate projections derived from four GCMs were analyzed (see Methods). The GWMs used a consistent river routine network map, in which all grid cells in each basin were connected. The models explicitly account for water management, including water withdrawals, reservoir operation, and land-use changes. Considering these direct human influences on the terrestrial water cycle is essential when studying drought in the Anthropocene;[42] in this study, these processes after 2006 were fixed at the 2005 level (2005soc) to discuss the impact of climate change while avoiding uncertainties stemming from socioeconomic changes, but their development during the historical period was incorporated.

Hydrological drought is defined as the condition when the daily river discharge is lower than or equal to a daily variable threshold for which seasonality is considered (see Methods). The drought detection process was performed for each grid cell at the daily scale. Then, the frequency of drought days (FDD; % of a season or a year) was estimated for each year, focusing on drought conditions lasting longer than one month. The results

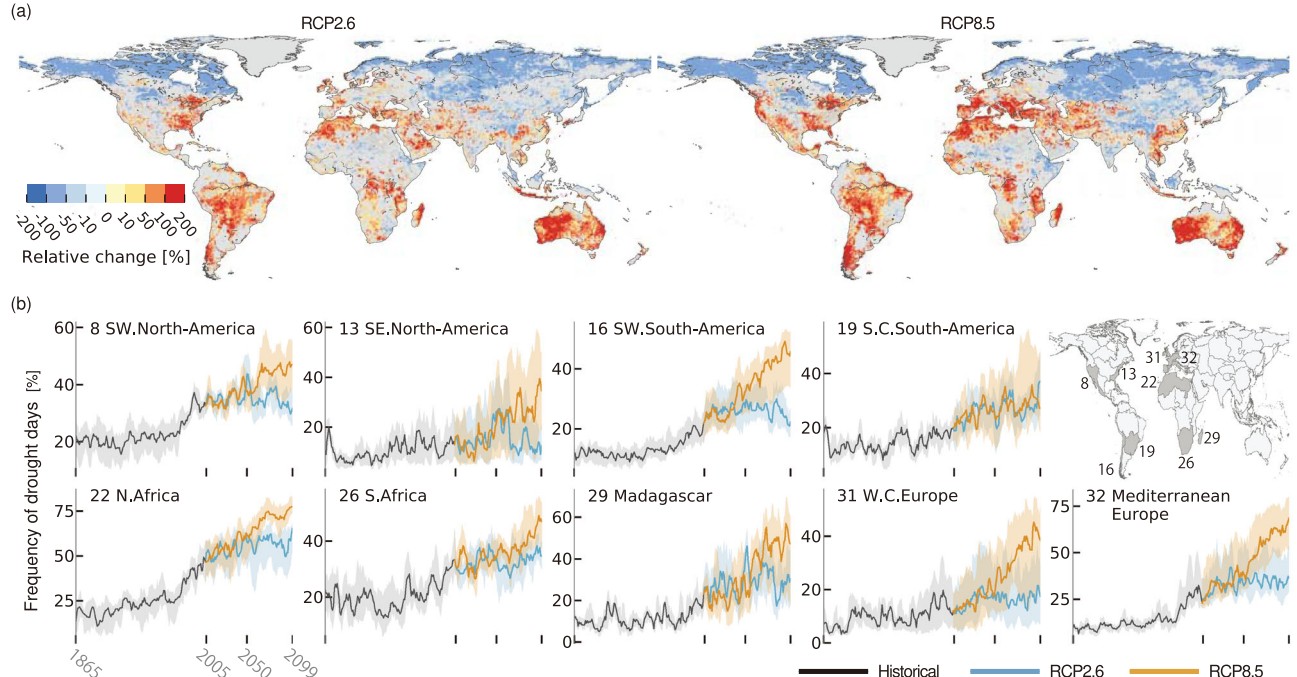

**Fig. 1 Projected spatiotemporal changes in the frequency of drought days (FDD) under climate change (during the low-flow season). a** The maps show the ensemble median values of the climatological percent changes derived for the FDDs in the mid-21st century (2036–2065) under RCP2.6 and RCP8.5 compared to the historical period (1971–2005). The results obtained for the low-flow season are presented. The colors indicate the direction and magnitude of the change [%]. Grids with nonsignificant changes between two periods (derived according to a two-sided Kolmogorov–Smirnov test (confidence level 0.05)) are marked with gray, and grids in which agreement in the sign of change among ensemble members is lower than 60% are also shown in gray (Supplementary Fig. 8). Additionally, Greenland is masked out in gray. **b** The plots present time series of the regional average FDDs derived during the low-flow season [%] from 1865 to 2099 under RCP2.6 and RCP8.5 in the nine selected regions. These regions show robust median TFE$_5$ values under either or both RCP2.6 or/and RCP8.5 (Fig. 2a). The lines present the ensemble median time series, and the shading shows the uncertainty in terms of the interquartile range across ensemble members. The region names and numbers are listed in Supplementary Fig. 3. The time series estimated for the rest of the regions are presented in Supplementary Fig. 5.

outlined in the following section are centered on changes in abnormal low-flow conditions over three months in the low-flow season (Supplementary Fig. 2; see Methods). Nonetheless, we also investigated abnormal low-flow conditions during the high-flow season and on the annual scale because a significant decrease in river discharge during the wet season could upset a sound annual hydrological cycle in a given region, causing the hydrological stationarity to collapse. Because the smaller the spatial scale is, the larger the influence of internal variability is[43], and because we assessed river discharge in this study, a time series analysis was carried out for 59 subcontinental regions delineated by river basin boundaries[44] (Supplementary Fig. 3).

**Projected spatial and temporal changes in drought conditions**. Regarding the low-flow season, the FDD is projected to increase significantly in 25% (28%) of the global land areas under RCP2.6 (RCP8.5) by the mid-21st century (2036–2065) compared to the values observed in the historical period (1971–2005) (statistically significant; two-sided Kolmogorov–Smirnov (KS) test; $p = 0.05$ and with >60% agreement among ensemble members) (Fig. 1a). In particular, 6% (9%) of the land areas are expected to experience a pronounced increase in FDD by the mid-21st century under RCP2.6 (RCP8.5), with the increased FDDs more than twofold. Under RCP2.6, the fraction of the land area in which pronounced FDD increases are expected is slightly higher (+1%) in the late-21st century (2070–2099) than in the mid-21st century, whereas the fraction projected under RCP8.5 is expected to increase significantly by the late-21st century (+10%) (Supplementary Fig. 4).

Overall, large FDD increases are projected in several drought-intensification hotspot regions, including the Mediterranean, Western and Central Europe, the Middle East, Central Asia, the United States, Central and southern South America, West to South Africa, and Australia. The time series of the regional average FDDs in these regions show that the drought condition frequencies have already increased compared to those recorded in the historical period and are projected to continue to increase (Fig. 1b; the time series for all 59 regions are presented in Supplementary Fig. 5). These increases also apply to the high-flow season in the hotspot regions; hence, crucially, abnormal low-flow conditions are expected to increase throughout the year (Supplementary Figs. 6 and 7). These large increases are predicted to be apparent by the mid-21st century under both climate scenarios. Importantly, the interensemble agreement on the sign of change and the signal-to-noise ratio regarding the ensemble member spread both indicate high confidence regarding the significant increases in the regions listed above (Supplementary Figs. 8 and 9; see Methods). The hotspots of intensified hydrological drought and the related statistics for these regions are comparable to the findings of previous drought-projection studies[30,45]. Precipitation is projected to decrease, and evapotranspiration is expected to increase due to the warmer climate conditions despite the decreased precipitation expected in some regions (Supplementary Fig. 10). Conversely, the FDD is projected to differ little or to decline in parts of Asia, Eastern Africa, and most regions in the northern high latitudes. Under RCP2.6, the difference between the mid- and late-21st century is modest. In several regions, while FDD increases over large spatial extents are projected by the mid-21st century, the diversions from

the historical level become statistically nonsignificant in the late-21st century (e.g., in the eastern United States, eastern Brazil, and southern South Africa). These trend reversals are consistent with the decreased precipitation expected until the mid-21st century followed by the increased precipitation projected by the late-21st century.

Consequently, we found substantial regional disparities in the impacts of climate change and the pace of changes among the 59 regions (Supplementary Fig. 5). Concerning the scenario differences, several regions exhibited increased regional average FDDs that were comparable between the RCPs until the 2040 s, in which time the differences in the GHG concentrations between the two scenarios were still relatively small (approximately 10–20%) (e.g., Fig. 1b). Thereafter, the regional average FDDs are predicted to stabilize or decrease under RCP2.6 due to the climate stabilization caused by the assumed decreases in GHG and aerosol concentrations. In contrast, the regional average FDDs are expected to continue to increase until the end of the 21st century under RCP8.5. Overall, these results indicate that in several regions, the future regional average FDD is expected to be larger than the maximum regional average FDD observed during the historical period. These regions could thus face unprecedented drought conditions within the coming decades due to the significant intensification of regional drought conditions.

**Timing of the first emergence of unprecedented drought**. Considering the irreducible uncertainty resulting from internal variabilities[11,19,46] in addition to the interensemble member spread, we estimated the TFE of unprecedented regional conditions from 100,000 resampled time series of regional average FDDs for each of 20 GCM and GWM combinations using the block-wise bootstrap method[41] assuming quadratic long-term trends and contingent natural variability (see Methods). The large bootstrap samples is composed of 2 million time-series in total for each region, enabling the robustness evaluation of the emergence. In addition, unlike preceding studies, this study presents the likelihood of TFE occurrence based on the cumulative probability over time every five years from 2010 until the end of the analysis period.

Regarding the case in which consecutive drought conditions occur equal to or for longer than five years ($TFE_5$) in terms of the low-flow season, $TFE_5$ was detected in 11 and 18 out of 59 regions, including the hotspot regions listed above, under RCP2.6 and RCP8.5, respectively (Fig. 2a). Similar to previous ToE studies, Fig. 2a presents the ensemble median values as representative TFE values. Note that the figure selectively shows regions in which the $TFE_5$ occurrence by 2100 is robust in terms of the median value (see Methods). Furthermore, 4 and 13 regions exhibited high $TFE_5$ probabilities (>66%) during the 21st century under RCP2.6 and RCP8.5, respectively, although the cumulative distribution functions (CDFs) showed an unignorable range of uncertainty in the time of emergence (Fig. 2b and Supplementary Fig. 11). These results indicate that the regions are likely to experience unprecedented regional drought conditions by the time under each considered scenario.

In particular, three regions show early $TFE_5$ under RCP8.5; the median $TFE_5$ values of Southwestern South America (SWS), Mediterranean Europe (MED), and Northern Africa (NAF) were approximately 2020, 2035, and 2040, respectively; moreover, $TFE_5$ was detected in 98%, 93%, and 67% of the large samples in these three regions, respectively, by 2050 (Fig. 2b and Supplementary Table 1). The sizes of the uncertainty ranges in cumulative probability at 2050 were below 2%, 5%, and 10% in SWS, MED, and NAF, respectively. Importantly, their $TFE_5$ occurrences during the 21st century were particularly robust

compared to those of other regions. In SWS, MED, and NAF, $TFE_5$ occurred by 2045, 2050, and 2080 in more than 95% of the resampled results, respectively (with uncertainty ranges of 2041–2046, 2049–2054, and 2075–2084, respectively) (Fig. 2b). These results indicate that the regional drought conditions in these regions are expected to shift to the higher-frequency side by the indicated time in response to global warming in addition to internal variabilities; additionally, with a high likelihood, these regions are very likely to experience unprecedented regional drought conditions under RCP8.5. All 20 original ensemble members, namely, the GCM and GWM combinations, consistently predict early $TFE_5$ in the three regions, although $TFE_5$ spreads were found among the original members in terms of other regions, especially due to the GCMs (Supplementary Fig. 12). On the other hand, the results obtained for RCP2.6 exhibit only one region in which more than 95% of the resampled results indicated $TFE_5$ by 2100 (SWS), while MED also showed a relatively high likelihood of $TFE_5$ occurrence compared to the other regions (Fig. 2b and Supplementary Fig. 11). Hence, the first emergence of unprecedented regional conditions by 2100 is particularly robust in SWS, followed by that in MED, regardless of the considered scenario.

The scenario differences shown in the CDFs demonstrate the importance of adopting emission pathways to prevent or delay these unprecedented conditions (Fig. 2b and Supplementary Fig. 11). Although the differences between the two scenarios are rather small in terms of the regions reflecting earlier median $TFE_5$ values (<2050, such as SWS), the discrepancies between scenarios are not very small (>20–30 years, e.g., in NAF), the uncertainty ranges do not overlap, or the median $TFE_5$ values are detected only under RCP8.5 during the middle or late stage of the 21st century (e.g., in West and Central Europe). These scenario differences are more evident when comparing the cumulative probabilities of $TFE_5$ at the end of the 21st century. Exceptionally, the probability derived under RCP2.6 is higher than that derived under RCP8.5 in a few regions, such as Southern Central South America. In general, the $TFE_5$ CDFs show that under RCP2.6, the growth rate of the likelihood of $TFE_5$ is originally low to modest; otherwise, the rate begins to decline before or in approximately 2050. Consequently, the cumulative probability grows at a slower rate in the second half of the 21st century. In contrast, the cumulative probability of $TFE_5$ under RCP8.5 was found to increase steadily over time unless it reached saturation. These differences in $TFE_5$ between the two considered scenarios are consistent with the trends presented in Fig. 1 and Supplementary Fig. 4. Emission pathways are, therefore, considered to critically influence drought conditions in these regions.

Despite the variabilities in the regional average FDDs at several-year scales, the impact of continued warming under RCP8.5 could lead to the occurrence of TFEs consecutively longer than five years in specific regions (Supplementary Figs. 13 and 14). With regard to the median TFE, for instance, $TFE_{10}$ for the low-flow season was detected in 13 regions under RCP8.5. Crucially, seven regions (SWS, MED, NAF, Southwestern and Southern North America, Madagascar, and Western and Central Europe) show median TFEs for consecutive exceedance until the end of the 21st century (>20–70 years) under RCP8.5. The longer an assumed consecutive exceedance is, the later the median TFE, i.e., the time at which the cumulative probability of TFE occurrence exceeds 0.5, is. Because the threshold of the TFE detection used in this analysis represents an extreme condition that occurred only once during the historical 145-year period, these TFEs reflecting longer consecutive exceedance durations indicate that the regions need to prepare for a new hydrological drought regime by the indicated time. In contrast, $TFE_{10}$ under RCP2.6 is exhibited only in SWS and MED, where a large

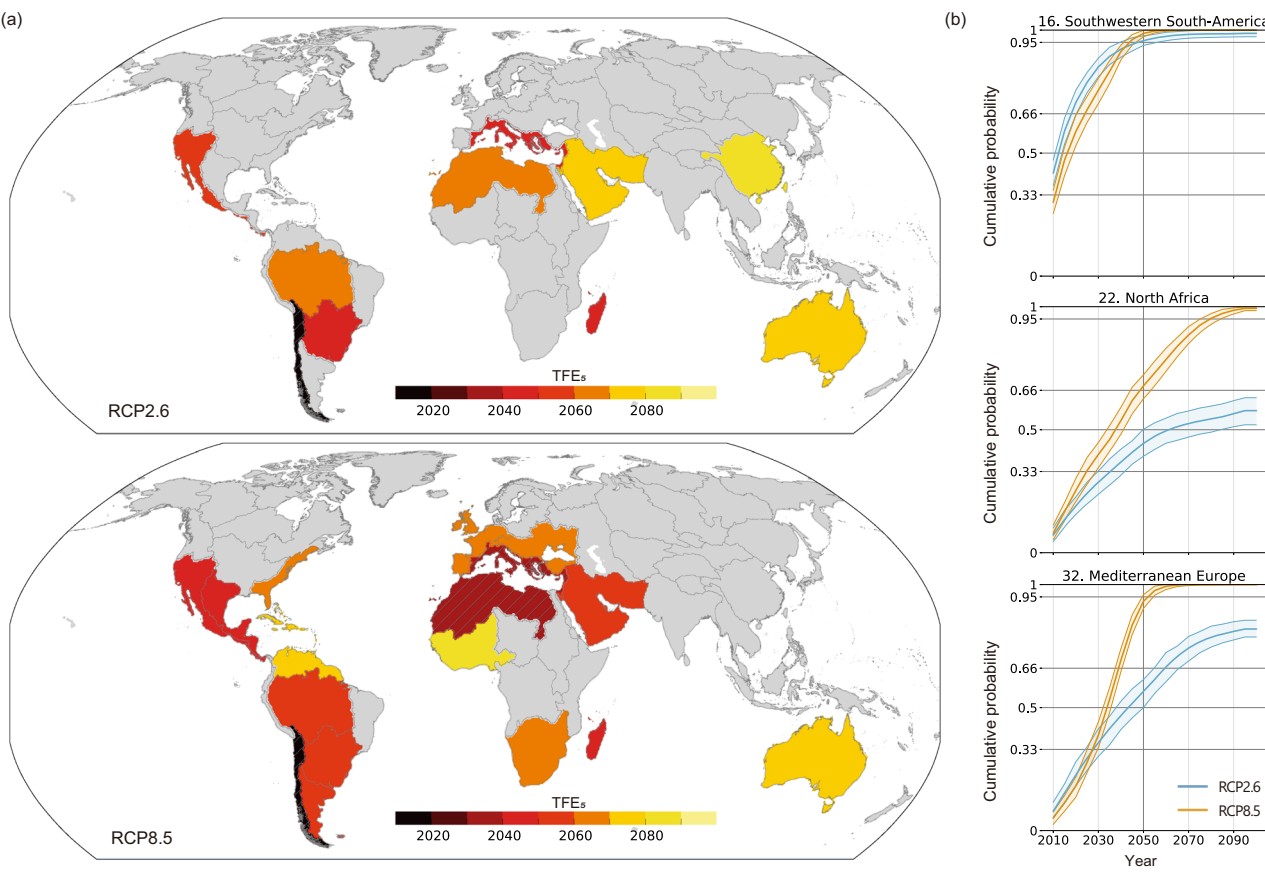

**Fig. 2 Timing of the first emergence (TFE) of consecutive unprecedented regional drought conditions (during the low-flow season). a** Timing of the first onset of consecutive exceedance for equal to or more than five years compared to the historical maximum value ($TFE_5$) under RCP2.6 and RCP8.5 in the 59 regions. The ensemble median results derived from the resampled time series are presented. Only regions in which the ensemble median $TFE_5$ obtained during the 21st century is statistically robust at the ±5% level by the bootstrap test are shown in color (see Methods). Otherwise, regions are shown in gray. The hatched areas indicate robust $TFE_5$ signals; in these regions, more than 95% of the bootstrap ensemble members showed $TFE_5$ during the 21st century. **b** The cumulative distribution functions (CDFs) of $TFE_5$ occurrences under the two considered RCPs as a function of time, i.e., the likelihood of $TFE_5$ occurrence over time, in three regions with particularly robust $TFE_5$ signals concerning RCP8.5. The CDFs shown as solid lines are estimated from the entire resampled results. Considering the internal variabilities and original ensemble member spreads, the shading represents the uncertainty in the cumulative probability of $TFE_5$ estimated from resampled ensemble member subsets (see Methods). The cumulative probabilities of $TFE_5$ occurrence by 2050 and by the end of the 21st century are given in Supplementary Table 1. The same CDF plots for all regions with the median $TFE_5$ shown in color in **a** are presented in Supplementary Fig. 11. The regional definitions were derived following the HydroBASINS level-2 product[44] (Supplementary Fig. 3).

regional average FDD is projected by the middle of the 21st century. Note that the regional average FDDs of other regions could exceed their maximum historical values even under RCP2.6, though the consecutive duration is shorter than the criterion defined.

Moreover, $TFE_5$ in the high-flow season and at the annual scale was also detected in similar regions (Supplementary Figs. 15 and 16). In general, because river discharge has clear seasonality, a sufficient surface water supply during the high-flow season is crucial for avoiding prolonged and aggravated drought conditions in the subsequent low-flow season. In the SWS, MED, NAF, and the Middle East regions, the $TFE_5$ occurrence in the high-flow season during the 21st century is robust under RCP8.5, with more than 95% of the large bootstrap members detecting $TFE_5$, and the median $TFE_5$ values estimated in these regions are earlier than those derived in other regions. SWS, MED, and the Middle East also show relatively early median $TFE_5$ values in RCP2.6. The regions in which $TFE_5$ is indicated during the 21st century are not necessarily consistent with those in which $TFE_5$ is identified during the low-flow season, and fewer and more regions show

median $TFE_5$ values during the 21st century under RCP2.6 and RCP8.5, respectively. The dry conditions that are expected to occur during low- and high-flow seasons are considered to be interrelated, and annual-scale warming impacts are more apparent in regions where $TFE_5$ was found in both seasons.

Overall, the results demonstrate that the choice of emission pathways leads to different $TFE_5$ likelihoods. More regions show robust $TFE_5$ values under RCP8.5 than under RCP2.6. The CDFs exhibit a higher likelihood under RCP8.5 in the long term (Supplementary Fig. 11). The difference between scenarios is more apparent in the second half of the 21st century in regions where drought is projected to intensify, although there are a few exceptions. Furthermore, the difference between the two climate scenarios is statistically significant, with a lower ensemble median derived under RCP2.6 than under RCP8.5 regarding the total number of years under unprecedented regional drought conditions during the 21st century, at 18 out of 59 regions with $TFE_5$ under either or both scenarios. These results also underscore the long-term benefits of the lower emission scenario (Fig. 3; see Methods).

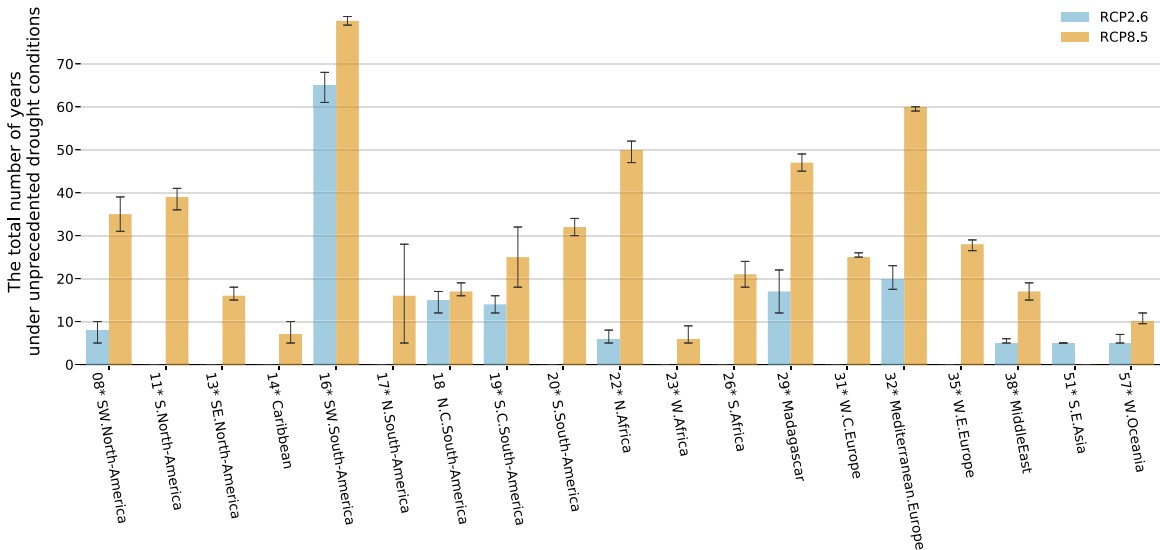

**Fig. 3 Total numbers of years in which unprecedented conditions are expected from 2010 to 2099.** The error bar shows the 5–95% confidence interval in terms of the ensemble medians (see Methods). The asterisks next to region names indicate that the difference between the two scenarios is statistically significant with regard to the median derived for that region.

Nonetheless, it should be stressed that achieving a low-emission pathway cannot fully avert the emergence of unprecedented regional drought conditions. Regional average FDDs have already increased in many regions, and global warming is projected to continue for a few decades under RCP2.6. Importantly, pronounced $TFE_5$ likelihoods are estimated in a few regions even under RCP2.6. The total number of years in which unprecedented regional drought conditions are expected in these regions is not small, even under the low-emission scenario. SWS shows the earliest $TFE_5$ with a high robustness under both RCPs, indicating that a potentially imminent and significant intensification of regional drought conditions is expected to occur regardless of the emission scenario. MED exhibits a relatively high $TFE_5$ likelihood by 2050 even under RCP2.6, and high likelihoods (>66%) by the end of the century are also found under both RCPs in North Central and South Central South America and in Madagascar (Supplementary Fig. 11). Although these regions have faced severe drought conditions during recent decades, regional drought conditions are likely to continue to be exacerbated over several decades in the future (Fig. 1b). Therefore, even if society immediately begins reducing GHG and aerosol emissions and continuously follows the guidelines established for RCP2.6, the stringent mitigation scenario, adequate preparedness involving effective adaptation measures is still essential in these regions.

The TFE analysis results lead to the following question: to what extent can the potential of the emergence of unprecedented drought conditions be reduced if the climate mitigation target of 1.5 °C or well below 2 °C is achieved? TFEs can be interpreted from the perspective of the global mean temperature rise ($\Delta$GMT) because while the four GCMs have different climate sensitivities, a correspondence exists between time and the temperature projections; this is called the global temperature of emergence[12,17] (see Methods). Concerning the transient condition expected under RCP8.5, Fig. 4 presents $\Delta$GMT values corresponding to the median $TFE_5$ values and CDFs as a function of $\Delta$GMT for the three regions in which robust $TFE_5$ values were found during the 21st century. The CDFs estimated for other regions are presented in Supplementary Fig. 17.

We find that several regions are likely to experience unprecedented regional drought conditions even if the $\Delta$GMT

value remains at a relatively low level. The median $TFE_5$ values in SWS and MED correspond to $\Delta$GMT values below and equal to the 2 °C level, and those in NAF and Madagascar occur at $\Delta$GMT values between 2 and 2.5 °C (Fig. 4a). However, the results also show that ensuring lower $\Delta$GMTs are essential for reducing the likelihood of $TFE_5$, suggesting that pursuing the Paris Agreement target should be critical. In all regions where median $TFE_5$ values were detected only under RCP8.5, the $\Delta$GMT values at the median $TFE_5$ are higher than 3 °C. Moreover, the CDFs can more quantitatively demonstrate the extent to which the likelihood differs among the $\Delta$GMTs (Fig. 4b and Supplementary Fig. 17). For instance, regarding SWS, MED, and NAF, more than 95% of the large ensemble members detected $TFE_5$ when the $\Delta$GMT values reached approximately 2.9 °C, 3.3 °C, and 4.2 °C, respectively (the sizes of the uncertainty ranges were 0.4 °C, 0.3 °C, and 0.5 °C, respectively). On the other hand, the $TFE_5$ likelihoods at the 2 °C $\Delta$GMT level were lower, at 76%, 52%, and 36% in SWS, MED, and NAF, respectively (with uncertainty range sizes of 10%, 10%, and 11%, respectively); these values were further reduced to 49%, 15%, and 19% at the 1.5 °C level (with uncertainty range sizes of 10%, 7%, and 9%). As a result, achieving the Paris Agreement target is considered to be effective for almost all regions to reduce the $TFE_5$ likelihoods to the level of unlikely (<33%).

## Discussion

This study is the first to report the time of the first emergence of unprecedented regional drought conditions with regard to river discharge at the global scale, building on the robust approaches established in previous studies. The drought timing analysis could be extended with different parameter sets or even different drought analysis types because, in general, both drought and ToE can be analyzed with various potential approaches. Regarding ToE, in particular, unprecedented conditions can be interpreted in multiple ways depending on the purpose of a given study. In terms of the minimum duration of consecutive exceedances, we demonstrated that there is not necessarily a linear relationship between the minimum duration and the TFE (Supplementary Figs. 13 and 14). Although most of the previous ToE studies mentioned above focused on the onset of permanent exceedances

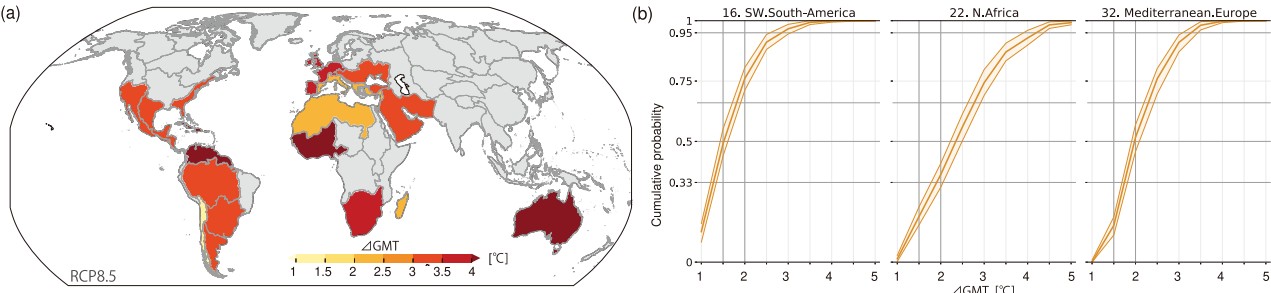

**Fig. 4 Global mean temperature rise corresponding to regional TFE5 values derived under RCP8.5. a** The global mean temperature rise (ΔGMTs [°C]) above the preindustrial level (1850–1900) corresponds to each regional median TFE5 presented in Fig. 2a. The ΔGMT values are derived from each GCM containing the resampled ensemble members that constitute the overall median TFE5. **b** Cumulative probability functions of TFE5 as a function of ΔGMT, corresponding to those presented in Fig. 2b. Note that the maximum ΔGMT derived under RCP8.5 is approximately 6.0 °C.

until the end of the 21st century, some studies considered shorter exceedances[13,36]. We also applied a shorter minimum duration while assuming that the studied systems are resilient to one- or two-year abnormal drought conditions but that abnormal drought conditions that have not occurred in recent decades and that last for multiple consecutive years may lead to irreversible changes and dire consequences[47]. For instance, the recent California drought lasted almost five years, causing severe damage to humans and natural systems[48–51]. However, despite a broad sense of drought and unprecedented conditions, elucidating the relationships between analysis parameters and sectoral damage remains challenging. As this study aims to provide the first TFE estimates of unprecedented regional drought conditions, further analyses of these relationships are left to future studies.

Our findings shed light on the early emergence of unprecedented regional conditions, but our estimates retain irreducible uncertainty due to internal variabilities and, in principle, reducible uncertainties stemming from the model structures, even though we used the state-of-art bias-corrected GCM projection dataset and the state-of-art GWMs. Considering these internal variabilities and the different climate sensitivities among GCMs, the use of a larger GCM set or large initial-condition ensembles from GCMs could improve the robustness of the TFE analysis. In terms of the internal variabilities, we evaluated the uncertainty and estimated the TFE likelihood using the bootstrap method; in contrast, previous ToE studies employed more GMCs to consider a wider range of these uncertainties or applied large initial-condition ensembles derived from one GCM to address the internal variabilities[20,52,53].

Likewise, GWMs also have inherent model biases. For instance, only two (LPJmL[54] and MATISRO[55,56]) and one (LPJmL) out of five GWMs include the stomatal response to $CO_2$ and vegetation dynamics, respectively. Both of these processes affect transpiration, but they could have contradictory functions in terrestrial hydrology;[57] studies[58,59] have shown that the weakened stomatal opening induced by increased $CO_2$ concentrations reduces transpiration and ameliorates hydrological drought risks, thus implying the likelihood of later TFEs than those estimated in this study. In contrast, other studies[60] have shown that due to increased $CO_2$ concentrations and consequential longer and warmer growing seasons, increased vegetation growth could result in increased evapotranspiration and reduced terrestrial water availability, suggesting earlier TFE of unprecedented drought. Our $CO_2$ experiment in which TFEs with variable $CO_2$ concentrations (default) and $CO_2$ concentrations fixed at the 2005 level were compared also showed certain differences under RCP8.5, although these differences were not apparent under RCP2.6 (Supplementary Fig. 18). We found earlier TFE5 in the fixed $CO_2$ results for several combinations of regions and

ensemble members regarding MATSIRO. Nevertheless, both earlier and later TFE5 were observed in response to the $CO_2$ differences in the LPJmL results due to the mixed impact of the processes described above.

Another important assumption made in GWM simulations, which is to avoid additional sources of uncertainty, is that the domestic and industrial water withdrawal, land use, and reservoir capacity were fixed at the 2005 level (2005soc). An increase in water withdrawal due to socioeconomic growth[61] can exacerbate stream drought[62], but it is still challenging to obtain robust projections or scenarios regarding these sectors. Nonetheless, the projected water withdrawal and land use data available for only RCP2.6 (rcp26soc) in the Inter-Sectoral Impact Model Intercomparison Project phase 2b (ISIMIP2b) enable the impact of these human activities on TFEs to be assessed (Supplementary Fig. 19). The results estimated with rcp26soc show earlier TFE5 or newly detected TFE5 values for some combinations of regions and ensemble members and, consequently, for the ensemble medians. Those scenarios for RCP8.5 (rcp85soc) are not available in ISIMIP2b, but it can be assumed that the impact of these human activities would be higher in rcp85soc than in rcp26soc because of the corresponding shared socioeconomic pathway (SSP) scenario (i.e., SSP5) and warmer climate conditions. Thus, in reality, TFE5 is thought to occur earlier than the TFE5 indicated by the 2005soc assumption discussed above due to increased water withdrawals. When comparing the results of the $CO_2$ and rcp26soc experiments, we find clearer differences in the latter. The contradictory functions of the stomatal responses and vegetation dynamics with regard to drought conditions and quantitative human activity scenarios are important research directions with which future studies can enable more robust drought projections. However, importantly, the differences found in these experiments regarding the GWM simulations fall within the range of the ensemble member spread, so further understanding of the structural differences in GCMs and GWMs is imperative.

We find that the regional average frequency of drought days (FDD) is anticipated to increase substantially in several regions even during the first half of the 21st century, indicating that the regional drought conditions are likely to shift toward more severe conditions. To define the time of first emergence (TFE) of unprecedented regional drought conditions, we investigated the time series of the regional average FDD under high- and low-emission scenarios to find the time at which the analyzed value exceeds the historical maximum value consecutively for a certain number of years. When discussing a consecutive exceedance duration equal to or longer than five years (TFE5) while considering uncertainty arising due to internal variabilities, the results highlight a distinct regional disparity in the warming impacts. The relatively early TFE5 are found to precede RCP2.6-

related climate stabilization. The robust median TFE$_5$ indicates that 11 and 18 out of 59 regions are expected to experience unprecedented regional drought conditions under RCP2.6 and RCP8.5, respectively (Fig. 2a). The TFE$_5$ occurrence during the 21st century is particularly robust in three regions under RCP8.5: SWS, MED, and NAF. The ensemble median TFE$_5$ values indicate that SWS and MED are expected to face TFE$_5$ conditions within the next 30 years with a likelihood of greater than 50% regardless of the considered RCP scenario. Reflected in the fewer regions with robust median TFE$_5$ values and the lower TFE$_5$ likelihood under RCP2.6, especially in the second half of the 21st century, the results demonstrate that strong mitigation efforts could potentially inhibit increases in the frequency of unprecedented drought conditions and effectively reduce the likelihood of TFE$_5$ occurrence. Furthermore, our estimates indicate that achieving the Paris Agreement target could avert unprecedented regional drought conditions in many regions. However, unprecedented regional drought conditions, in which the regional average FDD is larger than the maximum value in the past 145 years and this FDD exceedance lasts longer than five consecutive years, are projected to not be unlikely in some regions by the end of the century even under the low-emission scenario. Thus, appropriate and feasible adaptation plans are essential for overcoming the expected extraordinarily severe dry conditions. Our results shed light on potential concerns that the existing infrastructures and practices that were designed based on historical records or experiences may become insufficient in the near future to cope with droughts in a warmer climate in some specific regions. Therefore, it is crucial to improve our preparedness in the given time horizon before unprecedented drought conditions emerge.

## Methods

**Data, models, and simulation settings**. Century-long multimodel offline hydrological simulations conducted in the Inter-Sectoral Impact Model Intercomparison Project phase 2b (ISIMIP2b;[63] https://www.isimip.org/) were analyzed. The five GWMs included three global hydrological models: CWatM[64], H08[65], WaterGAP2;[66,67] one global land surface model: MATSIRO;[55,56] and one dynamic global vegetation model: LPJmL[54]. Although a larger number of GWMs participated in ISIMIP2b, we selected the five GWMs that (1) accounted for direct human impacts on hydrological processes (e.g., reservoir operation and water withdrawals for irrigation, domestic, and industrial water use) and (2) provided results for both RCP2.6 and RCP8.5. LPJmL and MATSIRO include the stomatal response to $CO_2$ concentrations. All simulations were conducted at 0.5°×0.5° spatial resolution, following the ISIMIP2b simulation protocol (https://www.isimip.org/protocol/#isimip2b). The models were forced by the ISIMIP2b bias-corrected and spatially downscaled (0.5°×0.5°) daily meteorological forcing data[68] derived from the simulations of four GCMs involved in the Coupled Model Intercomparison Project 5 (CMIP5): HadGEM2-ES[69], IPSL-CM5A-LR[70], GFDL-ESM2M[71], and MIROC5[72]. We note the caveat that the bias-corrected climate forcing data still retain model uncertainty for the future period resulting from climate sensitivity and internal variabilities[73]. The future projection period in line with the RCPs is 2006–2099. All GWMs applied a consistent flow direction map, DDM30;[74] this map assumes that each grid cell has a representative river and is connected to other grid cells to be organized into a drainage basin. The river discharge simulation skills of these GWMs were validated in preceding studies[75–77]. Other input datasets, such as reservoir locations and capacities and irrigation areas, varied over time during the historical period (1861–2005, histsoc) but were set to remain constant at the 2005 level during the future period (2006–2099, 2005soc). Irrigation water demand was, however, simulated by the GWMs in response to climate change.

**Drought definition and detection algorithm**. The daily moving window threshold method was applied to detect and quantify drought conditions[30,78,79]. A drought day was defined as a day in which the daily discharge was lower than or equal to a threshold for the day. The daily varying threshold was given as the X-percentile historical daily discharge ($Q_X$). This threshold was set at each grid cell for each day. To obtain $Q_X$ for a given day and grid cell, window sampling with a size of 31 days centered on the target day was applied using the data collected over the 145-year historical reference period (1861–2005), resulting in an X-percentile daily discharge estimated from 4495 samples (31 days × 145 years). Hence, this sampling method enables the consideration of the yearly variabilities and monthly characteristics of low-flow conditions during the historical 145 years centered on a given day of the year. Furthermore, because we considered day-to-day variabilities in addition to year-to-year variabilities when defining the $Q_X$ value, this approach enabled stricter

drought detection compared to those applied in preceding studies[80,81] that referred only to year-to-year variabilities on each day of the year and subsequently smoothed the time series of Q values (Supplementary Fig. 20). We used a consistent $Q_X$ for the future period (the nontransient threshold[7]) to estimate the overall long-term changes in comparison to the historical period.

Two other essential parameters were considered for detecting drought days, $\tau_x$ and $Len_x$ [days]; these parameters were the thresholds for short periods of nondrought days and drought days, respectively. The term $\tau_x$ denotes a shorter interruption than that of x-days between consecutive drought days in a drought period. Such short periods of nondrought days between lengthy periods of drought days were considered drought days in the drought period due to the consideration of the pooling effect[79]. $Len_x$ is the duration of a negligible drought period shorter than x-days, and such short periods of sequential drought days were not counted as drought days when estimating drought periods. In particular, the latter process enabled the analysis to focus on long (thus implying severe) drought events. This study focused on the results of the following set of parameters: $Q_{80}$, $\tau_4$, and $Len_{30}$. Considering that we analyzed multiple models at the global scale in this study, overall, these values were set in the ranges of each parameter used in preceding studies[79,82–89]. Then, drought days were calculated as the occurrence frequency in a given season or in a year, i.e., the frequency of drought days (FDD; % of the season or the year). The low- and high-flow seasons considered the three months during which the average river discharge during the historical period was the lowest and highest in one year, respectively; these seasons were set for each grid cell and ensemble member (Supplementary Fig. 2). Consequently, the FDDs calculated for each grid cell over the 239 years of study (1861–2099) were assessed in this study.

**Regional drought characteristics**. To define regional unprecedented drought conditions, this study explored the temporal evolution of regional average FDDs. In the time series analysis, we grouped the global land area into 59 subcontinental scale regions following the delineations established in HydroBASINS (level 2)[44] (Supplementary Fig. 3). The dataset divides the global land area into nine continents (level 1) and further splits each continent into up to 9 large subunits (level 2) based on river basins. A finer-scale assessment (e.g., a grid or a finer basin scale) would be more useful for practical applications, but the results of this study are presented at the subcontinental scale because, considering the uncertainty that arises due to internal variabilities in GCM projections, spatially aggregated information derived with large-scale sampling is recommended to improve the statistical robustness when investigating changes in extreme events[43,90]. The smaller the scale of analysis is, the greater the uncertainty resulting from internal variability effects is[30]. For each year, the regional average FDD was estimated from all grid cells in a given region except grid cells containing glacial ice. The latitudinal grid area differences were considered by applying area weights when calculating the regional average FDDs. Moreover, considering the temporal uncertainties associated with the internal variabilities in the GCM projections, the regional average FDDs were serially estimated for each year with 5-year sampling windows that ended at each year, i.e., samples during 1861–1865 constituted the regional average FDD obtained for 1865. This window-sampling method filtered out interannual variabilities while multidecadal variabilities were retained. Thus, using the drought assessment for 1861–2099, the time series of the regional average FDDs during 1865–2099 was investigated.

**Timing of the first emergence (TFE) of unprecedented drought**. The regional average FDDs were analyzed to identify pronounced departures from the historical variability ranges. Distinct from several ToE studies that applied the signal-to-noise ratio[6,17], Kolmogorov–Smirnov test[11,16,18], probability ratio[25], or Hellinger distance metric[21], in this study, we applied the method proposed by Mora et al.[9] to estimate the point in time at which regional drought characteristics deviated significantly from past regional drought characteristics. We identified the first year when the regional average FDD time series exceeded the corresponding maximum value during the historical baseline period (1865–2005) and subsequently remained beyond this historical range for a certain number of consecutive years afterward (Supplementary Fig. 1). Unlike the quasi-preindustrial baseline period[12,24], the baseline period covered the entire historical period until 2005, considering recent increases in regional average FDDs and uncertainties arising due to internal variabilities. As the simulation period was until 2099, no TFE was detected if the consecutive departure did not appear by a certain number of years before the end of the simulation period; i.e., for TFE$_5$, any exceedances identified during the 2095–2099 period were eliminated. We then estimated the TFE for each member of a large dataset that was resampled (see the next section), and the ensemble median was presented as a representative result. Considering several recently observed drought events that lasted 2–6 years (e.g., the Southern African drought in 2018; the Brazilian drought[91] in 2014–2017; and the California drought in 2011–2017[92,93]), we focused on cases with a first consecutive period lasting longer than five years in this study, i.e., TFE$_5$ (see Discussion). For instance, Trisos et al. 2020[13], who performed a ToE study on biodiversity loss, also investigated exceedances lasing at least five years, although almost all other ToE studies have focused on permanent exceedance. The results derived for different minimum exceedance durations are presented in Supplementary Figs. 13 and 14.

**The block-wise bootstrap method and evaluation of robustness in TFE$_5$.** We evaluated the temporal uncertainty in TFE arising from internal variabilities and model uncertainties by the block-wise bootstrap resampling method[41]. Only four bias-corrected climate projections are available in ISIMIP2b, unlike previous ToE studies in which more GCM projections or large initial-condition ensemble data provided by a single GCM were used. Our uncertainty evaluation can be summarized by the following procedure: (1) For each ensemble member and region, the regional average FDD time series during 1985–2099 was decomposed into a trend and a series of anomalies. The trend was estimated consistently as a quadratic function in all cases (Supplementary Figs. 21 and 22). (2) Considering the serial dependence of the time series that stems from several-year-scale natural variability cycles, we applied a five-year nonoverlapping block-wise resampling method. We sought the effective decorrelation time with which the autocorrelation of the time series becomes less than 0.3 and found that the 5-year-lag autocorrelations were lower than 0.3 for most of the time series of ensemble members and regions (Supplementary Figs. 23 and 24). A few ensemble members for specific regions, such as parts of Asia and the Middle East, show higher values than others, but their correlations are also not strong with the log size. (3) For each case, 100,000 random resamplings were performed to develop a large-resampled time series. Hence, for each region with 20 ensemble members, we estimated 2 million TFE samples in total. (4) Then, considering the uncertainty that arose due to internal climate variabilities and the model structures, the TFE spread derived from the large-resampled time series was used as a measure of the uncertainty in the TFE results.

We estimated the TFE likelihood over time at a 5-year interval using the large TFE samples and presented the results as cumulative probabilities as a function of time (Fig. 2b and Supplementary Fig. 11). In addition, the derived cumulative distribution functions (CDFs) demonstrated the 5–95% confidence intervals of TFE, indicating the uncertainty in the TFE results. The uncertainty range of each CDF was estimated using numerous subsets of the large ensemble data by using another bootstrap method. First, two million samples were randomly shuffled and grouped into 2000 subsets; then, a CDF was estimated for each subset. The minimum and maximum ranges of these 2000 CDFs provided the uncertainty range of the main CDF. Furthermore, the robustness of the TFEs was evaluated from three perspectives in this study. (i) If more than 95% of the samples in the large ensemble exhibited TFEs by the end of the 21st century, the TFE occurrence during the century was considered robust (very likely).; (ii) the robustness of the median TFE was evaluated using the same method as that used to obtain the CDF uncertainty range, as explained above. If more than 95% of the median TFEs derived from the 2000 subsets reflect TFEs during the 21st century, the median TFE$_5$ during the century was considered robust. Importantly, the uncertainty range of the CDF indicates that the spread of the median TFE$_5$ values was not extensively large compared to the overall TFE spread (i.e., the 5–95% confidence interval).; (iii) If no more than 33% of samples showed TFE at a given time, the likelihood of TFE occurrence by this time is unlikely following the definition established by the Intergovernmental Panel on Climate Change (IPCC).

**Statistical test of the total number of years under unprecedented drought conditions.** As opposed to TFE$_5$ which focuses only on the onset timing of the first emergence of drought conditions, we also estimated the total number of years at which the regional average FDD was larger than a given threshold during 2010–2099 (Fig. 3). Note that, similar to TFE$_5$, short exceedances lasting less than five years were not counted. Statistical tests examining the differences between scenarios were carried out in terms of the median values. An approach similar to that used to estimate the uncertainty range of the CDF was applied. Two thousand median value samples were calculated for each scenario, and 2000 samples of their differences were also calculated. Assuming a 5–95% confidence interval, if the lowest 5th-percentile value among the large samples of differences was larger than 0 or if the highest 5th-percentile value was less than 0, the difference between the two sample groups was considered to be statistically significant.

**Global mean temperature rise (ΔGMT).** TFE can be converted into ΔGMT, but the relationships between TFE and ΔGMT vary among GCMs due to the different climate sensitivities of the models. For each GCM, the yearly GMT was calculated as the 31-year climatology centered on the analyzed year; then, the ΔGMT value, which is equal to the difference between the average GMT during the preindustrial period (1850–1900) and each yearly GMT, was calculated. When the TFE year was later than 2084, the ΔGMT value obtained for 2084 was considered. When calculating ΔGMT using the large-ensemble-median TFEs, the ensemble mean of the ΔGMTs derived from all corresponding members was used.

**Ensemble statistics.** Throughout this paper, the ensemble median results derived from the 20 ensemble members or the larger resampled ensemble members were presented for each analysis as representative outcomes; we obtained results regarding the spatial and temporal changes in the drought frequency (Fig. 1), the TFE analysis (Fig. 2), and estimations of the total unprecedented years (Fig. 3b). First, these analyses were carried out for each ensemble member; then, the ensemble median results of each analysis were presented. The model uncertainty associated with the impacts of climate change shown in Fig. 1 was evaluated using two metrics: the ensemble member agreement and the signal-to-noise ratio

(Supplementary Figs. 9 and 10). The former metric reflects the degree of agreement regarding the sign of changes across all ensemble members. Assuming the noise is the ensemble member spread reflected in the climatological changes, the signal-to-noise ratio was calculated as the ensemble median of the impacts of climate change divided by the interquartile range of the climate change impacts among ensemble members[30,45]. For the time series of the regional average FDDs (Fig. 1 and Supplementary Fig. 5), the model uncertainties were presented as interquartile ranges derived across the ensemble member results.

## Data availability

The minimum dataset generated in this study has been available in DataON (https://dataon.kisti.re.kr/, https://doi.org/10.22711/idr/938). With this dataset, the resampled large ensemble data can be reconstructed by the scripts available from the Github repository described below. The original hydrological simulation results are available from the ISIMIP project portal (https://www.isimip.org/outputdata/, https://doi.org/10.5880/PIK.2020.004).

## Code availability

All processed data and figures in this study were generated using Python. The relevant portions of the Python scripts used to process the results and develop the graphic presentation[94] are available from our GitHub and Zenodo repositories (https://github.com/yusuke61/tfe_scripts, https://doi.org/10.5281/zenodo.6488507).

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

## Acknowledgements

This research is supported by the "Integrated Research Program for Advancing Climate Models (TOUGOU Program) (JPMXD0717935457, JPMXD0717935715)" sponsored by the Ministry of Education, Culture, Sports, Science, and Technology (MEXT) of Japan. Decarbonized and Sustainable Society Research Program and Climate Change Adaptation Research Programs in National Institute for Environmental Studies also supported this study. We would like to thank the ISMIP (https://www.isimip.org) project for supplying the data and the modeling community for carrying out these crucial simulations. MATSIRO simulations were performed on SGI UV20 at the National Institute for Environmental Studies. Y.P. acknowledges the support from the National Science Foundation (CAREER Award, grant #: 1752729). H.K. was supported by a National Research Foundation of Korea (NRF) grant funded by the Korean Government (MSIT) (2021H1D3A2A03097768 and 2018R1A5A7025409).

## Author contributions

Y.S., K.Y., and T.O. conceived the research. Y.S., H.K., and H.S. designed the detail of the analysis and developed the algorithm. Y.S., T.Y., Y.P., P.B., N.H., J.E.S.B., H.M.S., D.G., and S.O. conducted hydrological simulations under the ISIMIP2b project. Y.S. processed the simulation results, conducted the analyses, and prepared the graphics. Y.S. interpreted the results and prepared the draft with significant contributions from Y.P., H.S., T.Y., N.H., and Y.W. K.Y., H.K., P.B., E.B., H.M.S., D.G., S.O., S.N.G., J.E.S.B., and T.O. discussed the outcome and made substantial contribution in editing the manuscript.

## Competing interests

The authors declare no competing interests.
