## [Peer Review File · Nature Communications]

The timing of unprecedented hydrological drought under climate changeReviewers' Comments:

Reviewer #1:

Remarks to the Author:

Review for "The timing of unprecedented hydrological drought under climate change" by Satoh et al. for Nature Communications

The authors present an analysis of the time of emergence of global hydrological drought using a climate and hydrologic modeling framework. They assess the time at which hydrologic drought exceeds the historic variability under two climate scenarios and also assess the impact on potentially exposed population to unprecedented drought conditions. They find that even under the most stringent anthropogenic forcing scenario (RCP 2.6), an initial emergence of hydrologic drought is expected by mid 21st century in many parts of the globe.

While I think using river discharge estimates to quantify drought globally is necessary to better understand the anthropogenic impact on drought, and the population exposure to drought is important in order to quantify the true impact of drought emergence on a region, I do have some major points that the authors need to address.

Firstly, the time of first emergence (TFE5) can be a bit misleading for a wider audience. This is because while a signal can emerge for five years, it could be due to the changing magnitude of internal variability. Consider ENSO as a cause of climate variability in a specific region (e.g., Western North America) which has a 4-7 year cycle. If ENSO variability, or the effects of ENSO are amplified, we might see a TFE5 in Western North America, but we could also see a similar intensified wet period in the following years. This is especially significant for water resources management, as the authors state in their introduction (lines 43-48) – it is unclear if one should plan for only increasingly dry conditions or highly variable climate conditions. One suggestion is to include results for TFE > 5 years as well as permanent emergence, and/or to include the actual number of years that the TFE5 occurs.

Additionally, this issue is amplified because the authors rely on a relatively short historic period (30 years) and a single simulation of each GCM-GWM model combination to quantify historical variability of drought conditions, which could also be highly influenced by climate variability especially on regional scales. While this can actually lead to conservative estimates of ToE, it can lead to some less conservative ones as well (see Hawkins et al., 2014). To overcome this issue, I suggest using a longer historic or pre-industrial period if available in the ISIMPIb2 simulations and use a moving window to calculate the time of emergence, as in Abatzoglou et al. (2019). (Ideally, one would use multiple large ensemble simulations to quantify the historic variability, however, I understand that these GCM-GWM simulations are not available and would take a lot of effort/time to run!)

Moreover, Mankin et al. (2017), specifically show the influence of internal variability on estimating the exposure of human population to hydroclimate changes. Given the similarity of the research questions posed by the authors of this study, this study should be cited and discussed by the authors in order to address the additional insight that could be gained if a large ensemble of GCM-GWM simulations did exist. In fact, Mankin et al. (2017) show that "irreducible uncertainty" stemming from internal variability alone causes 90% of the population to reside in locations where sub-annual water deficits could either increase or decrease. This is important for this study to put their findings in Figure 3, and the differences between SSPs/RCPs in perspective.

Additionally, I have some line by line comments that the authors need to address.

Line 57-61: Starting this sentence with "However,..." is a bit confusing, especially with the additional clause at the end. I think the sentence needs to be restructured/split up.

Additionally, could the authors clarify why not using bias correction is an issue in the Touma et al. study? Bias correction parameters are usually established in the historic period and then propagated

to the future period. In Touma et al., there was a direct comparison between historic and future climates in each GCM, and not compared to observations. Is bias correction still needed in this case?

Line 59: Please correct "special extent" to "spatial extent".

Lines 83-84 & 288-289: I think it is important to clarify in the main part of the text (and methods) how river discharge is considered on a grid point scale (I am guessing that not all grid points have a river running through them?)

Lines 131-132: I think the inference that the small differences in the RCP2.6 and 8.5 scenarios are due specifically to the small differences in GHG concentrations is not supported by this analysis. In addition to GHG, aerosols are considerably different in the different scenarios and could also cause these changes (see Westervelt et al., 2015). Additionally, if this is the direct cause, the authors need to discuss the mechanisms that link changes in GHG to changes in FDD for the broader readership of Nature Communications.

Figure 2: Please state why you only show these 8 regions per RCP in panel b.

Line 155: Please state the names of regions 18 and 17 in the text.

Lines 166-169: As previously noted, the authors' inference that GHG mitigation alone will lead to mitigation of these impacts on drought is unsupported by this analysis. If this is indeed the main solution, the authors need to discuss the mechanisms that link changes in GHG to changes in FDD for the broader readership of Nature Communications.

References:

- Abatzoglou, J. T., Williams, A. P., & Barbero, R. (2019). Global Emergence of Anthropogenic Climate Change in Fire Weather Indices. *Geophysical Research Letters*, 46(1), 326–336. <https://doi.org/10.1029/2018GL080959>
- Hawkins, E., Anderson, B., Diffenbaugh, N. S., Mahlstein, I., Betts, R. A., Hegerl, G., et al. (2014). Uncertainties in the timing of unprecedented climates. *Nature*, 511(7507), E3-5. <https://doi.org/10.1038/nature13523>
- Mankin, J. S., Viviroli, D., Singh, D., Giuliani, M., Anghileri, D., & Castelletti, A. (n.d.). Influence of internal variability on population exposure to hydroclimatic changes.
- Westervelt, D. M., Horowitz, L. W., Naik, V., Golaz, J. C., & Mauzerall, D. L. (2015). Radiative forcing and climate response to projected 21st century aerosol decreases. *Atmospheric Chemistry and Physics*, 15(22), 12681–12703. <https://doi.org/10.5194/acp-15-12681-2015>

Reviewer #2:

Remarks to the Author:

The topic examined here by Satoh et al., on the timings of the emergence of unprecedented hydrological drought – that crosses historical benchmark – is a very relevant one since such extreme events have significant environmental, ecological and socio-economical consequences. As such the authors make a claim that "... unprecedented conditions could be unavoidable even under the well-below 2°C warming target of the Paris Agreement" and that six of the analyzed regions "...would face unprecedented conditions within the next 20 years regardless of the GHG emissions". These are quite strong claims with implications on adaptation planning and mitigation strategies (and that too on a rather short-term); and therefore the underlying analyses require greater scrutiny.

1. Modeling issues

All analyses conducted in this study are based on the "offline hydrological simulations" from five global water models (GWMs) forced with (four) GCMs data. Here I see a potential issue: several recent

studies (e.g., Mily and Dunne 2016; Yang et al., 2019) have pointed out issues with use of offline models in exaggerating water-deficit (drought) conditions, for example due to missing factors either in forcing or process representation that account for CO₂ fertilization effects or dynamic (vegetation) feedback loops.

Upon further reading I do find only one of the five GWMs used in this study have a dynamic global vegetation-modeling component (LPJmL). How different are the results (changes in FDD and timings of the emergence of unprecedented hydrological drought) based on this model compared to others? If this model appears to be an outlier – then clearly any multi-model ensemble study would put a less weight on results of such a model compared to others (i.e., 4 vs 1). Here, I would like to recall the study by Prudhomme et al., (2014) on the projections/uncertainty of future hydrologic droughts – conducted in the earlier phase of the same ISIMIP project and indeed some of this work's co-authors were also part of the Prudhomme et al. work. Their study clearly show a large divergence among the model projections for future droughts – and especially due to models that account for the dynamic response of plants to CO₂ and climate, which simulated little or no increase in drought frequency. It was therefore recommended to use the diverse range of impact models to better capture the uncertainty and arrive to robust conclusions. And to my reading this is something important that has been missing from this study.

It is therefore remained an open question to what degree such missing (forcings or vegetation feedback loops) factors play a role in author's investigation? Are there findings of the near-future occurrence (in the next 20-years) of the emergence of unprecedented hydrological drought, reliable or robust enough in light of these missing components?

Yang, Y., Roderick, M. L., Zhang, S., McVicar, T. R., & Donohue, R. J. (2019). Hydrologic implications of vegetation response to elevated CO₂ in climate projections. *Nature Climate Change*, 9(1), 44-48.

Milly, P. C. D. & Dunne, K. A. Potential evapotranspiration and continental drying. *Nat. Clim. Change* 6, 946–949 (2016).

Swann, A. L. S., Hoffman, F. M., Koven, C. D. & Randerson, J. T. Plant responses to increasing CO₂ reduce estimates of climate impacts on drought severity. *Proc. Natl Acad. Sci. USA* 113, 10019–10024 (2016).

Prudhomme, Christel, et al. "Hydrological droughts in the 21st century, hotspots and uncertainties from a global multimodel ensemble experiment." *Proceedings of the National Academy of Sciences* 111.9 (2014): 3262-3267.

2. (Missing) role of internal/natural climate variability:

The authors' analyses are based on a very limited sample of forcings (four GCMs with a single realization) with no consideration for internal/natural climate variability – which is crucial for analyzing the timing of emergence (e.g., Hawkins & Sutton, 2012; Deser et al., 2012; Hawkins et al. 2014; Lehner et al. 2017). As demonstrated in a comprehensive study by Lehner et al. 2020 (also see earlier study by Hawkins and Sutton, 2019), the role of internal (climate) variability is seemingly important and dominant across several climatic regions, compared to other uncertainty sources (emission scenarios or model structural uncertainty) especially at a short time horizon. Not considering this aspect in the analysis can possibly make an overconfident statement on the timing of emergence (considering the uncertainty around the estimates) and the conclusion drawn in this study that "if society begins reducing greenhouse gas emissions immediately and continuously to follow the stringent mitigation scenario, the initial emergences of such unprecedented conditions would occur before or around the beginning of climate stabilization (the 2040-2050s), calling for urgent adaptation measures."

Hawkins, E., & Sutton, R. (2012). Time of emergence of climate signals. *Geophysical Research Letters*, 39(1).

Hawkins, Ed, et al. "Uncertainties in the timing of unprecedented climates." *Nature* 511.7507 (2014): E3-E5.

Deser, C., Phillips, A., Bourdette, V., & Teng, H. (2012). Uncertainty in climate change projections:

the role of internal variability. *Climate dynamics*, 38(3-4), 527-546.

Lehner, F., Deser, C., & Terray, L. (2017). Toward a new estimate of "time of emergence" of anthropogenic warming: Insights from dynamical adjustment and a large initial-condition model ensemble. *Journal of Climate*, 30(19), 7739-7756.

Hawkins, E., & Sutton, R. (2009). The potential to narrow uncertainty in regional climate predictions. *Bulletin of the American Meteorological Society*, 90(8), 1095-1108.

Lehner, F., Deser, C., Maher, N., Marotzke, J., Fischer, E. M., Brunner, L., Knutti, R., and Hawkins, E.: Partitioning climate projection uncertainty with multiple large ensembles and CMIP5/6, *Earth Syst. Dynam.*, 11, 491-508,

3. Novelty and concerns on some of the key arguments put forward in the Introductory paragraphs:

The idea for analyzing the time of emergence of unprecedented events has been around for quite some time and reported in several studies (some of them being also appropriately cited in this study). So on this point I do not find this study providing any methodological advancements. Specifically the authors themselves have pointed out the earlier study by Touma et al. on the timing of the emergence of first unprecedented future drought globally. The authors then make a point that Touma et al. used local runoff data and that "it is also crucial to consider direct human influence on the terrestrial water cycle (e.g., water withdrawal, irrigation and reservoir operation) that cannot be ignored in the context of drought in the Anthropocene" and that "... they employed future projections from General Circulation Models (GCMs) without bias-correction ..."

On these points they argue the use of 5 GWMs that accounted for direct human impacts: such impacts were considered temporally variable only during the historical period (1979-2005, *histsoc*) but were set to remain constant at the 2005 level during the future period (2006-2099, *2005soc*) other than irrigation water demand that was simulated by the GWMs in response to climate change. So essentially the authors mainly consider the impact of varying climate conditions in the future period (and no changes in direct human influences).

Fair enough - my point here why to make a strong argument for the consideration of human influences in the Introductory paragraph - in comparison to previous study - and then latter reading through the Methods we have to find that such considerations were not accounted especially during the future period - for which this study makes all sort of conclusions (changes in FDD, timing of emergence, etc). The same thing goes for the argument of bias-correction in the sense that there is no systematic analysis conducted by the authors to point out that bias-correction would really change their conclusions.

4. Use of SREX regions:

I have difficulties in understanding the use of SREX regions by the authors - especially when they argued in favor of using the river discharge, instead of local runoff. The SREX regions do not respect the hydrological boundary of a river basin. The differences in analysis of TFE5 are very much evident when aggregating results to either SREX regions or to basin-country scale (comparing results shown in Fig. 2 and Supplementary Fig. 8). For example look at the median TFE5 value in SREX region 13 (from Fig. 2 it is around 2040; whereas in Fig 8 it is highly variable between 2050-2080; if I read it right). Anyways the authors need to provide a convincing argument regarding the selection of the regional domain.

5. Time-periods:

Their need to be a consistency in the time-period used in the study: At Line 80 it states - "... we investigate hydrological drought globally from 1979 to 2099 under RCP2.6 and RCP8.5"; whereas in the caption of Fig. 1 it states, "... compared to a historical period (1971-2005)".

6. Model validations:

Lines 287-88: I could not find in the three cited references 48-50 (note that one of them is published in a conference proceeding), the model validation exercise focusing on FDD - which is subject of this

study.

7. Regional drought characteristics:

Line 333. It is not clear what the authors mean by "...yearly regional drought PDFs were serially estimated with a 5-year sampling windows that ended in each year".

Reviewer #3:

Remarks to the Author:

Review of "The timing of unprecedented hydrological drought under climate change" by Satoh et al.

A. Main comments

This manuscript present estimates of the Time of Emergence of unprecedented hydrological droughts for regional areas at the global scale. Although the motivation and even some results in some extent are valuable, the methodology chosen for the present study have 3 major issues that would in my opinion prevent its publication in Nature Communications. These 3 points are summarized below and further detailed in specific comments.

A.1. Definition of hydrological drought

The authors argue that their study advance the state of the art of hydrological drought projection at the global scale as they base their analysis on streamflow and not runoff as done in the literature. I cannot agree more on this point with authors. Unfortunately, they firstly chose to define hydrological drought by anomalies with respect to the annual average cycle, allowing for example to consider lower-than-average high flows in the high-flow season to be considered a a drought period. I believe that the motivation of using streamflow comes from a will to better approach real-life behaviour and issues, but this choice actually draw things away from actual drought and low-flow management. Moreover, the set-up used in ISIMIP2b has everything related to water management fixed at the level of year 2005, which make things again away from plausible futures. I have strong reserves on selling such a study with underlying assumptions that could be interpreted in the wrong way. Indeed, the assumption of no change in irrigation area, reservoir operation, and so on (L293-296) prevent one to have a full view of what may happen in terms of hydrological drought as simulated by GHMs that can simulated (and that's a real advance in my view) the anthropogenic water cycle (with abstractions, reservoir operations, and land-use change).

This whole issue may be solved in (1) specifying right from the abstract the underlying assumptions in hydrological projections, and (2) specifying right from the abstract what kind of hydrological drought is studied here, or rather looking at low-flows than streamflow anomalies.

A.2. Natural variability and TFE

The choice made here by authors to define "unprecedented" is based on exceeding the record value simulated over a historical period, following the choice made by Mora et al. (2013). But the difference between the two studies is that Mora et al. considered the record value over a 145-year period (1860-2005), when the present study considers a record value over a 34-year period (1979-2013). One criticism expressed by Hawkins et al. (2014) on the study by Mora et al. is that they may find a too early emergence of the signal due to their methodological choices. This is evident that the present study would largely fall under the same criticism, as hydrological drought variability cannot be estimated from a so short time period. Depending on the interannual and mutidecadal variability expressed in each of the 4 selected GCMs here, exceeding the record value of the 1979-2013, and even for several consecutive years, may statistically happen as early as the 2010s even in a fully stationary climate. This is already a strong point, but my experience in bias-correction showed that the interannual variability from a corrected simulation often increases outside of the reference period

used for calibrating the bias-correction. This would add to the issue described above. As a conclusion to this point, and for an individual GCM projection, the choice made for defining “unprecedented” is highly biased towards and early emergence of the signal.

This issue may be solved in considering a longer reference period (if available in simulations) or considering another threshold for considering expulsion from historical variability.

A.3. Internal variability and TFE

The second main comment is relative to the estimation of future variability and the associated uncertainty, which is quite related to the robustness of TFE values. It is maybe not a coincidence that not that many studies looked at the Time of Emergence of unprecedented droughts at the global scale based on offline runoff/streamflow simulations (as pointed out by the authors), because available ensembles do not have the required size (actually the required size of driving GCM runs) that would allow a robust assessment of such features that are highly dependent on the representation of internal variability. Authors recognize in the discussion (in half a sentence L251) that a larger set of GCMs would be beneficial to the robustness of their findings. The number of driving GCMs is here 4, to be compared to 15, used in a study the authors refer to frequently in the manuscript (Touma et al., 2015). Results from Touma et al. are therefore undoubtedly more robust (statistically speaking only, as there are also other methodological choices that come into play) than the ones presented here that rely only on the temporal pattern of only 4 time series. Works by Deser et al. (2012a,b) clearly showed many years ago the importance of considering large ensembles from a single model (SME) to properly assess future trends. This is all the more important when estimating threshold-dependent features like Times of Emergence. Usually large SME are not available and one take multi-model ensemble (MMEs) instead, hoping that internal variability will be sampled through the use of many single runs. Note that methods currently aim at better describing this GCM internal variability even based on a limited number of runs (see e.g. Evin et al., 2019; Hingray et al., 2019). The TFE of the hydrological signal will be in the present study much too influenced by the actual realization from each of the 4 individual projections to deliver a robust estimate. As consequence, as shown by Fig. 2, the median TFE cannot represent the main mode of the distribution and sometimes actually falls between two modes corresponding to very distant times.

This issue is difficult to solve given the state of global offline simulations that are restricted to a limited number of driving GCM runs. One way forward may lie in looking at innovative recent techniques for estimating internal variability.

B. Specific comments

L49-51: The literature is indeed scarce in terms of time of emergence for drought-related variables. However, it would be worth mentioning that Vidal et al. (2016) explored the timing of emergence of low-flow conditions outside the natural (internal) hydroclimate variability at the catchment scale for both annual and 30-year averages of low-flow statistics. Pohl et al. (2020) also recently proposed a methodology to better estimate the time of emergence for streamflow variables, that would be worth exploring and at least mentioning.

L62-63: Bias-correction has limitations and drawbacks, and may lead to inconsistent changes when applied to future projections, as recently demonstrated by Chen et al. (2020). This should definitely be acknowledged in the manuscript.

L79-83: It is of high importance for the readers to understand that even if the authors look at future streamflow time series, these projections rely on the highly questionable (in terms of correspondence to plausible futures) hypothesis that reservoir location, reservoir operations, and irrigation area will remain at their 2005 level throughout all the 21st century! This part of the ISIMPI2b protocol is specified in the Methods section, but it would deserve a front row notification to the readers.

L83-85: Moreover, it is also of high value for the reader to understand here that what is considered as a “hydrological drought” here is a deviation from the average annual regime. That means that lower high flows during the high-flow season will be considered as a drought period and therefore contribute (of course depending on the parameters defined in the Methods section) to the frequency of drought days. This is due to the approach chosen here, based on a daily variable threshold which identifies daily anomalies irrespective of the magnitude of streamflow, as shown recently by Stahl et al. (2020). Of course, this is something that does not reflect what is actually relevant in terms of drought management in the real life. This comment and the preceding one deeply influence the lecture of the results from this study and this should be strongly acknowledged, starting up from the abstract.

L93-106, Fig. 1: The 2D color scale is obviously perceptually misleading (and not appropriate for colorblind readers), and this issue has been voiced by many studies now (Staufer et al., 2015), and even in Nature (Hawkins, 2015). Kaye et al. (2012) even provided appropriate 2D colour scales for climate variables. Please reconsider the colour scales of Fig. 1, but also Extended Fig. 2, Supplementary Fig. 2, Supplementary Fig. 3, Supplementary Fig. 6, Supplementary Fig. 7, and in a lesser extent Supplementary Fig. 8.

L108-121: The ISIMIP fast-track experiments allowed a few years ago studies on the multimodel evolution of hydrological droughts and these should be acknowledged in this manuscript (Prudhomme et al., 2014, Giuntoli et al., 2015). Indeed, these studies also provide for example maps of the signal-to-noise ratio of the evolution of runoff drought (with a daily variable and fixed threshold respectively) that should be compared with the ones provided by the authors, notably in terms of hot spots.

L128-130: Actually, this precise statistical analysis may lead to similar results even in a fully stationary climate (even if the global homogeneity in significance would be highly improbable). Please consider presenting results from an alternative test set-up or justify the present one better.

L148-149, “The ensemble spread...” I guess there could be an overestimation or underestimation of the probability value with KDE given the censored time interval (2010-2100). Please provide some assessment of this limitation. Additionally, I don’t understand why the left time boundary is 2010 and not 2006 (start of the RCPs) or 2014 (outside of the bias-correction and reference period). Please provide some justification or modify it in the manuscript.

L142-153, Fig. 1: This is where the lack of consideration of internal variability in future projections actually emerges in the manuscript... The highly multimodal distribution shown in Fig. 1 for the TFE5 is actually driven by the sole 4 individual time series of the four driving GCMs of the ensemble considered here (4 GCMs x 5 GHMs/LSMs). This means that the internal variability is represented by only 4 series (for a given RCP) whereas it is well known for almost a decade that understanding the internal variability requires much more realizations (GCM runs) to grasp 21st century evolution and trends (Deser et al., 2012a,b). This is all the more important here that the Time of Emergence is highly dependent on this internal variability through the interannual and multidecadal variability of streamflow. Fig. 1 clearly shows the lack of robustness of TFE5 estimates. Were all 5 GHMs have similar performance in representing hydrological droughts, results would show only 4 distinct values derived from the 4 GCM forcing time series (for a given RCP). And using another run from each of the 4 GCMs considered would have led to quite different estimates, which is undoubtedly detrimental to the validity of results presented here. This is discussed in much more detail in the specific comments below.

L161-164: Authors recognize here the large inter-ensemble spread in TFE estimate. Considering both RCPs during the first half of the 21st century indeed increase the robustness of TFE estimates that fall before 2050s. But Fig. 1 actually considers the two RCPs separately (rightly so).

L166-169: To be clear, the two above comments do not imply that I disagree with what is stated here.

My point is that the statistical analysis as presented here is not robust enough to fully support it. For example, considering the interquartile range from the ensemble comes down roughly to consider 2 GCMs out of 4...

L170-175: Results commented in this paragraph are the ones that are the less robust here given all comments above.

L176-184, Fig. 3: The error bars displayed in the left hand side clearly show the lack of robustness of the signal. This (non-)robustness could be assessed by comparing distributions under both RCPs by e.g. the Mann-kendall two-sided test, and would indeed rarely show any statistical difference across regions (or only by chance given the differences in interannual and multidecadal variability between RCP2.6 and RCP8.5 from any given GCM). Additionally, the light gray error bars are not visible enough on this graph, minimizing their visual impact...

L211-224: Given all my above comments on the lack of robustness of estimates of both the historical variability and future internal variability, I am quite reserved on these conclusions on the correspondence of TFEs with DeltaGMT. Supplementary Fig. 4 clearly shows this lack of robustness on temperature and TFE5.

L22-224: "Extraordinary" is a term used here as a synonym to "not experienced during the reference period". And this reference period is only 34-year long, which is quite short for understanding hydrological drought variability as a function of natural climate variability (see e.g. Caillouet et al., 2017). So I guess "extraordinary" is not the appropriate term here. As a consequence, as discussed in another comment below, finding unprecedented conditions could perfectly happen even in a complete stationary climate. There are many works on the probability of breaking historical records in the literature, and I would recommend getting deeper in these, all the more given the lack of robustness in this study of estimates of both historical natural variability and future natural (internal) variability.

L310-321: I wonder why such pooling technique has been selected here as the Sequent Peak Algorithm (Tallaksen et al., 1997) is much less parameterized and more easily understandable.

L334: Considering Q80 as a threshold does not correspond to identifying an "extreme phenomenon" as stated in the manuscript, as 73 days per year in average should be identified as dry days (less so with tau and len parameters, I agree, but still).

C. Technical corrections

L98: Fig. 1, caption: "significant level" → "confidence level"

L113: "profound increase" ?

L146, "shading" → shaded

L331: "grids" → "grid cells"

L352: "periods" → "period"

D. References

Caillouet, L., Vidal, J.-P., Sauquet, E., Devers, A., Graff, B. (2017) Ensemble reconstruction of spatio-temporal extreme low-flow events in France since 1871. *Hydrology and Earth System Sciences*, 21, 2923-2951. doi: 10.5194/hess-21-2923-2017

Chen, J., Brissette, F., Caya, D. (2020) Remaining error sources in bias-corrected climate model

outputs. *Climatic Change*, 162, 563-582. doi: 10.1007/s10584-020-02744-z

Deser, C., Knutti, R., Solomon, S., Phillips, A. S. (2012a) Communication of the role of natural variability in future North American climate. *Nature Climate Change*, 2, 775-779. doi: 10.1038/nclimate1562

Deser, C., Phillips, A., Bourdette, V., Teng, H. (2012b) Uncertainty in climate change projections: the role of internal variability. *Climate Dynamics*, 38, 527-546. doi: 10.1007/s00382-010-0977-x

Evin, G., Hingray, B., Blanchet, J., Eckert, N., Morin, S., Verfaillie, D. (2019) Partitioning uncertainty components of an incomplete ensemble of climate projections using data augmentation *Journal of Climate*, 32, 2423-2440. doi: 10.1175/JCLI-D-18-0606.1

Giuntoli, I., Vidal, J.-P., Prudhomme, C., Hannah, D. M. (2015) Future hydrological extremes: the uncertainty from multiple global climate and global hydrological models. *Earth System Dynamics*, 6, 267-285. doi: 10.5194/esd-6-267-2015

Hawkins, E., Anderson, B., Diffenbaugh, N., Mahlstein, I., Betts, R., Hegerl, G., Joshi, M., Knutti, R., McNeall, D., Solomon, S., Sutton, R., Syktus, J., Vecchi, G. (2014) Uncertainties in the timing of unprecedented climates. *Nature*, 511, E3-E5. doi: 10.1038/nature1352

Hawkins, E. (2015) Graphics: Scrap rainbow colour scales. *Nature*, 519, 291. doi: 10.1038/519291d

Hingray, B., Blanchet, J., Evin, G., Vidal, J.-P. (2019) Uncertainty component estimates in transient climate projections. *Climate Dynamics*, 53, 2501-2516. doi: 10.1007/s00382-019-04635-1

Kaye, N. R., Hartley, A., Hemming, D. (2012) Mapping the climate: guidance on appropriate techniques to map climate variables and their uncertainty. *Geoscientific Model Development*, 5, 245-256. doi: 10.5194/gmd-5-245-2012

Mora, C., Frazier, A. G., Longman, R. J., Dacks, R. S., Walton, M. M., Tong, E. J., Sanchez, J. J., Kaiser, L. R., Stender, Y. O., Anderson, J. M., Ambrosino, C. M., Fernandez-Silva, I., Giuseffi, L. M., Giambelluca, T. W. (2013) The projected timing of climate departure from recent variability. *Nature*, 502, 183-187. doi: 10.1038/nature12540

Pohl, E., Grenier, C., Vrac, M., Kageyama, M. (2020) Emerging climate signals in the Lena River catchment: a non-parametric statistical approach. *Hydrology and Earth System Sciences*, 24, 2817-2839. doi: 10.5194/hess-24-2817-2020

Prudhomme, C., Giuntoli, I., Robinson, E. L., Clark, D. B., Arnell, N. W., Dankers, R., Fekete, B. M., Franssen, W., Gerten, D., Gosling, S. N., Hagemann, S., Hannah, D. M., Kim, H., Masaki, Y., Satoh, Y., Stacke, T., Wada, Y., Wisser, D. (2014) Hydrological droughts in the 21st century, hotspots and uncertainties from a global multimodel ensemble experiment. *Proceedings of the National Academy of Sciences*, 111, 3262-3267. doi: 10.1073/pnas.1222473110

Stahl, K., Vidal, J.-P., Hannaford, J., Tjeldeman, E., Laaha, G., Gauster, T., Tallaksen, L. (2020) The challenges of hydrological drought definition, quantification and communication: an interdisciplinary perspective. *Proceedings of the International Association of Hydrological Sciences*, 383, 291-295. doi: 10.5194/piahs-383-291-2020

Stauffer, R., Mayr, G. J., Dabernig, M., Zeileis, A. (2015) Somewhere over the rainbow: How to make effective use of colors in meteorological visualizations. *Bulletin of the American Meteorological Society*, 96, 203-216. doi: 10.1175/BAMS-D-13-00155.1

Tallaksen, L. M., Madsen, H., Clausen, B. (1997) On the definition and modelling of streamflow drought duration and deficit volume. *Hydrological Sciences Journal*, 42, 15-33. doi: 10.1080/02626669709492003

Touma, D., Ashfaq, M., Nayak, M. A., Kao, S.-C., Diffenbaugh, N. S. (2015) A multi-model and multi-index evaluation of drought characteristics in the 21st century. *Journal of Hydrology*, 526, 196-207. doi: 10.1016/j.jhydrol.2014.12.011

Vidal, J.-P., Hingray, B., Magand, C., Sauquet, E., Ducharne, A. (2016) Hierarchy of climate and hydrological uncertainties in transient low-flow projections. *Hydrology and Earth System Sciences*, 20, 3651-3672. doi: 10.5194/hess-20-3651-2016

Reviewer #1 (Remarks to the Author):

The authors present an analysis of the time of emergence of global hydrological drought using a climate and hydrologic modeling framework. They assess the time at which hydrologic drought exceeds the historic variability under two climate scenarios and also assess the impact on potentially exposed population to unprecedented drought conditions. They find that even under the most stringent anthropogenic forcing scenario (RCP 2.6), an initial emergence of hydrologic drought is expected by the mid 21st century in many parts of the globe. While I think using river discharge estimates to quantify drought globally is necessary to better understand the anthropogenic impact on drought, and the population exposure to drought is important in order to quantify the true impact of drought emergence on a region, I do have some major points that the authors need to address.

We appreciate the time and effort the reviewer has dedicated to reaching a deliberate decision and to providing insightful feedback about our study. Thank you for providing us with another opportunity to improve our study. We revisited our study considering the reviewer's comments and have made substantial revisions in response to the reviewer's feedback. Major changes we made include:

1. While the uncertainty in TFE had been evaluated only by ensemble model spread in the previous version, we now apply a **bootstrap method to consider uncertainty stemming from internal variability** as well as model uncertainty (see Methods). The results are presented for only regions where ensemble median TFE₅ occurring during the 21st century is statistically robust with $\pm 5\%$ level by the bootstrap test. Our discussion is centered on more robust TFE₅ occurring during the 21st century, with equal to or more than 95% of bootstrap members show TFE₅ during the 21st century (see Methods). Because of this improvement, the main message of this study was also updated.
2. We **extend the historical reference period** from 35 years to 145 years (1861-2005) to better take into account a wider range of uncertainty due to internal variability.

Our point-by-point answers are listed below.

Firstly, the time of first emergence (TFE₅) can be a bit misleading for a wider audience. This is because while a signal can emerge for five years, it could be due to the changing magnitude of internal variability. Consider ENSO as a cause of climate variability in a specific region (e.g., Western North America) which has a 4-7 year cycle. If ENSO variability, or the effects of ENSO

are amplified, we might see a TFE5 in Western North America, but we could also see a similar intensified wet period in the following years. This is especially significant for water resources management, as the authors state in their introduction (lines 43-48) – it is unclear if one should plan for only increasingly dry conditions or highly variable climate conditions. One suggestion is to include results for TFE > 5 years as well as permanent emergence, and/or to include the actual number of years that the TFE5 occurs.

We thank the reviewer for the essential comments. In particular, we agree with the importance of internal variability in estimating the time of the first emergence of unprecedented regional drought conditions. In consideration of the possibility of amplified internal variability under climate change, Supplementary Figs. 9 and 10 present TFE_x (x=1, 4, 5, 6, 15, 20, permanent exceedance until end of the analysis period), and lines 241-252 describe that several regions exhibit unprecedented regional drought conditions longer than five years, in particular under RCP8.5. Additionally, the total number of years under unprecedented conditions during 2010-2099 is also presented in Fig. 3 and is described in lines 273-275. Furthermore, to overcome the limited number of bias-corrected GCM projections to consider uncertainty due to internal variability, we applied the block-wise bootstrap method to estimate TFEs (see Methods). This new approach enables the probabilistic representation of TFE, and we can evaluate uncertainty in TFEs considering internal variability. Then, we focus on robust TFEs (i.e., more than 95% of the large bootstrap samples show TFE, or ensemble median TFE₅ is statistically significant with $\pm 5\%$ confidence level) in the main text. Uncertainty ranges due to the internal variability and model spread are also presented based on the bootstrap results. As a result of the updated analysis, TFE₅ became later in several regions than the previous version. However, we still find robust and relatively early TFE₅. The conclusions and outcomes of the study were revised accordingly.

Additionally, this issue is amplified because the authors rely on a relatively short historic period (30 years) and a single simulation of each GCM-GWM model combination to quantify historical variability of drought conditions, which could also be highly influenced by climate variability especially on regional scales. While this can actually lead to conservative estimates of ToE, it can lead to some less conservative ones as well (see Hawkins et al., 2014). To overcome this issue, I suggest using a longer historic or pre-industrial period if available in the ISIMIP2b simulations and use a moving window to calculate the time of emergence, as in Abatzoglou et al. (2019).

(Ideally, one would use multiple large ensemble simulations to quantify the historic variability, however, I understand that these GCM-GWM simulations are not available and would take a lot of effort/time to run!)

Thank you for this useful suggestion. Following this comment, we extended the historical reference period from 1979-2013 (35 years) to 1861-2005 (145 years), which is the full length of the historical period in ISIMIP2b (Lines 431, 463-466, 473-475). Thus, the upper bound of

the historical internal variability, which is the threshold of TFE detection, is estimated from four times as many samples as the results in the previous manuscript. Additionally, the moving window approach was used to estimate the regional average FDD (Lines 463-466). This window sampling filters out interannual variability while retaining multidecadal variability.

In addition, please note that two more particular cares were taken for the internal variability when estimating TFE. First, rather than investigating smaller scales, such as the grid cell scale or the basin scale, we selected the subcontinental scale to improve the robustness of TFE (Lines 114-117, 454-459). It is known that the smaller the scale is, the larger the influence of internal variability. Additionally, preceding studies have indicated that spatially aggregated information with large-scale sampling is recommended to improve statistical robustness when investigating changes in extreme events^{1,2}. Second, the constraint, “for consecutively for a certain number of years”, aims to eliminate an episodic short exceedance due to large internal variability.

Moreover, Mankin et al. (2017), specifically show the influence of internal variability on estimating the exposure of human population to hydroclimate changes. Given the similarity of the research questions posed by the authors of this study, this study should be cited and discussed by the authors in order to address the additional insight that could be gained if a large ensemble of GCM-GWM simulations did exist. In fact, Mankin et al. (2017) show that “irreducible uncertainty” stemming from internal variability alone causes 90% of the population to reside in locations where sub-annual water deficits could either increase or decrease. This is important for this study to put their findings in Figure 3, and the differences between SSPs/RCPs in perspective.

We acknowledge the importance of large initial-condition ensemble simulations for improving the robustness of the TFE estimate. In addition to a large number of GCMs, hence, the importance of the initial-condition ensemble for a robust TFE estimation was added to the Discussion (line. 341-345). Mankin et al. (2017)³ is referred to in the lines. The block-wise bootstrap method is an alternative approach to these large ensemble data set. Also, we focus on robust TFE in the main text. Thus, given the results of Mankin et al. (2017) that a large population resides in regions with high uncertainty due to internal variability, we removed our result about potential populations exposed to unprecedented regional drought conditions.

line comments:

Line 57-61: Starting this sentence with "However,..." is a bit confusing, especially with the additional clause at the end. I think the sentence needs to be restructured/split up.

We revisited the introduction, and these sentences were also modified.

Additionally, could the authors clarify why not using bias correction is an issue in the Touma et al. study? Bias correction parameters are usually established in the historic period and then propagated to the future period. In Touma et al., there was a direct comparison between historic and future climates in each GCM, and not compared to observations. Is bias correction still needed in this case?

We assume that applying bias correction is better than using raw GCM output for climate change impact assessment because the robustness of atmospheric variables is guaranteed at a certain level, at least, during the historical period. We suppose the reviewer means that the impact of climate change (i.e., the difference between historical and future) evaluated by assessments with and without bias correction would be equal despite systematic bias during the historical period. However, for instance, Haerter et al. (2011)⁴ show that future changes in temperature from bias-corrected simulations can be different from those calculated using raw climate model simulations. Additionally, Johnson and Sharma (2015)⁵ investigated the impact of bias correction of GCM precipitation on future precipitation drought in Australia. They demonstrated that bias correction tended to moderate increases in future drought frequencies compared to raw GCM and, in some instances, improved the consensus amongst a large set of GCMs. These results indicate that the impact of climate change can be different between analyses with and without bias correction.

Furthermore, the impact of bias in atmospheric variables on terrestrial hydrological processes, such as runoff and evapotranspiration, could be even critical because the bias in atmospheric variables can result in another bias in terrestrial hydrology. Hence, the bias-corrected climate forcing data should play an important role in hydrological simulations and, consequently, drought assessments. That is why the ISIMIP community made efforts for offline simulations with bias-corrected climate forcing data.

Nonetheless, while we still mention it as a difference, we no longer stress these technical differences as issues of Touma *et al.* in the introductory paragraph. We instead stress the more fundamental questions: the importance of (1) the low emission scenario in view of the Paris agreement and (2) river discharge drought considering water resource management.

Line 59: Please correct “special extent” to “spatial extent”.

This typo was corrected.

Lines 83-84 & 288-289: I think it is important to clarify in the main part of the text (and methods) how river discharge is considered on a grid point scale (I am guessing that not all grid points have a river running through them?)

Following the ISIMIP2b simulation protocol, all GWMs use a consistent river routing network map, DDM30⁶. For each grid cell, this data set provides one surface water drainage direction. Contiguous grid cells in a basin are connected to each other by their respective drainage direction so that they organize into a drainage basin. Therefore, our large-scale GWMs used in this study assume that each grid cell has one representative river that is connected to up- and downstream.

We additionally described this information in the main text (Line 103) and Methods (Line 419-421).

Lines 131-132: I think the inference that the small differences in the RCP2.6 and 8.5 scenarios are due specifically to the small differences in GHG concentrations is not supported by this analysis. In addition to GHG, aerosols are considerably different in the different scenarios and could also cause these changes (see Westervelt et al., 2015). Additionally, if this is the direct cause, the authors need to discuss the mechanisms that link changes in GHG to changes in FDD for the broader readership of Nature Communications.

Thank you for this comment. We admit that it is premature to determine the causal connection between small differences in the increases in FDD and CO₂. We modified the sentence just to mention that the difference in CO₂ concentrations between RCP26 and RCP85 is 10-20% during the 2040s (Line 164-166). In addition, we now take aerosols into account as one of components of the RCP scenarios (Lines 167, Line 287).

On the other hand, we find that the effect of a decrease in aerosols on hydrological drought entails uncertainty. As a decrease in aerosols in general leads to an increase in both precipitation and evapotranspiration, it has the potential to both exacerbate and mitigate drought. Unfortunately, we cannot differentiate the impact of GHG and aerosols in this experimental setup and it is not our scope. Thus, we leave this issue for future studies. Instead, the changes in precipitation and evapotranspiration are presented in Supplementary Figure 7 and described in Lines 154-156 to understand the hydroclimatic background of the change in FDD under two emission scenarios.

Figure 2: Please state why you only show these 8 regions per RCP in panel b.

The revised Figure 2b now presents four regions that exhibit robust TFE₅ during the 21st century under either RCP2.6 or RCP8.5 because we focus on robust results. This is written in the figure caption in Figure 2b.

Line 155: Please state the names of regions 18 and 17 in the text.

Thank you for this comment. We agree with the importance of clearly stating the names of the regions. However, at the same time, we would like to keep the main text as simple as possible. Therefore, instead of the main text, we added names of the regions in the figure caption of Figure 2. (see the main text: Lines 226-227, and the figure caption in Fig. 2: Lines 188-195)

Lines 166-169: As previously noted, the authors' inference that GHG mitigation alone will lead to mitigation of these impacts on drought is unsupported by this analysis. If this is indeed the main solution, the authors need to discuss the mechanisms that link changes in GHG to changes in FDD for the broader readership of Nature Communications.

We acknowledge that highlighting only GHG mitigation is misleading. Thus, we mention the contribution of a decrease in aerosol emissions (Lines 167, Line 287). We are just saying that even under the strict mitigation scenario, unprecedented conditions would occur.

Reference:

1. Fischer, E. M., Beyerle, U. & Knutti, R. Robust spatially aggregated projections of climate extremes. *Nat. Clim. Chang.* **3**, 1033–1038 (2013).
2. Fischer, E. M. & Knutti, R. Detection of spatially aggregated changes in temperature and precipitation extremes. *Geomorphology* 547–554 (2014).
doi:10.1002/2013GL058499.Received
3. Mankin, J. S. *et al.* Influence of internal variability on population exposure to hydroclimatic changes. *Environ. Res. Lett.* **12**, (2017).
4. Haerter, J. O., Hagemann, S., Moseley, C. & Piani, C. Climate model bias correction and the role of timescales. *Hydrol. Earth Syst. Sci.* **15**, 1065–1079 (2011).
5. Johnson, F. & Sharma, A. What are the impacts of bias correction on future drought projections? *J. Hydrol.* **525**, 472–485 (2015).
6. Döll, P. & Lehner, B. Validation of a new global 30-min drainage direction map. *J. Hydrol.* **258**, 214–231 (2002).

Reviewer #2 (Remarks to the Author):

The topic examined here by Satoh et al., on the timings of the emergence of unprecedented hydrological drought – that crosses historical benchmark – is a very relevant one since such extreme events have significant environmental, ecological and socio-economical consequences. As such the authors make a claim that “... unprecedented conditions could be unavoidable even under the well-below 2°C warming target of the Paris Agreement” and that six of the analyzed regions “...would face unprecedented conditions within the next 20 years regardless of the GHG emissions”. These are quite strong claims with implications on adaptation planning and mitigation strategies (and that too on a rather short-term); and therefore the underlying analyses require greater scrutiny.

We appreciate the time and effort the reviewer has dedicated to reaching a deliberate decision and to providing insightful feedback about our study. Thank you for providing us with another opportunity to improve our study. We revisited our study considering the reviewer comments and have made substantial revisions in response to reviewer’s feedback. Major changes we made include:

1. Sensitivity analyses of **the CO₂ fertilization effect** and **the scenario of anthropogenic hydrological activities** were carried out.
2. While the uncertainty in TFE had been evaluated only by ensemble model spread in the previous version, we now apply **a bootstrap method to consider uncertainty stemming from internal variability** as well as model uncertainty (see Methods). The results are presented for only regions where ensemble median TFE₅ occurring during the 21st century is statistically robust with ±5% level by the bootstrap test. Our discussion is centered on more robust TFE₅ occurring during the 21st century, with equal to or more than 95% of bootstrap members show TFE₅ during the 21st century (see Methods). Because of this improvement, the main message of this study was also updated.
3. We **extend the historical reference period** from 35 years to 145 years (1861-2005) to better take into account a wider range of uncertainty due to internal variability.
4. **The introductory paragraphs** were largely revised to describe important assumptions of this study and highlight the fundamental scopes of this assessment.
5. **The regional definition** was changed from the SREX definition to the **AR6 definition**.

Our point-by-point answers are listed below.

1. Modeling issues

All analyses conducted in this study are based on the “offline hydrological simulations” from five global water models (GWMs) forced with (four) GCMs data. Here I see a potential issue: several recent studies (e.g., Mily and Dunee 2016; Yang et al., 2019) have pointed out issues with use of offline models in exaggerating water-deficit (drought) conditions, for example due to missing factors either in forcing or process representation that account for CO₂ fertilization effects or dynamic (vegetation) feedback loops.

Upon further reading I do find only one of the five GWMs used in this study have a dynamic global vegetation-modeling component (LPJmL). How different are the results (changes in FDD and timings of the emergence of unprecedented hydrological drought) based on this model compared to others? If this model appears to be an outlier – then clearly any multi-model ensemble study would put a less weight on results of such a model compared to others (i.e., 4 vs 1). Here, I would like to recall the study by Prudhomme et al., (2014) on the projections/uncertainty of future hydrologic droughts – conducted in the earlier phase of the same ISIMIP project and indeed some of this work’s co-authors were also part of the Prudhomme et al. work. Their study clearly shows a large divergence among the model projections for future droughts – and especially due to models that account for the dynamic response of plants to CO₂ and climate, which simulated little or no increase in drought frequency.

It was therefore recommended to use the diverse range of impact models to better capture the uncertainty and arrive to robust conclusions. And to my reading this is something important that has been missing from this study.

It is therefore remained an open question to what degree such missing (forcings or vegetation feedback loops) factors play a role in author’s investigation? Are there findings of the near-future occurrence (in the next 20-years) of the emergence of unprecedented hydrological drought, reliable or robust enough in light of these missing components?

We appreciate that the reviewer carefully went through the manuscript and provided important comments and suggestions. We agree that it is crucial to better understand the importance of the CO₂ fertilization effect and dynamic vegetation processes (DVPs), i.e., how much the results of GWMs with and without these processes differ compared to other model structural diversities. Therefore, we performed an extra sensitivity test on the impact of increased CO₂ and compared the TFE₅ of individual ensemble members to discuss how the TFE₅ of LPJmL and MATSIRO differ with and without CO₂ concentration increase and how they are different from others (Lines 346-358). As a result, we concluded that the lack of CO₂ fertilization effect and DVPs in a couple of GWMs in this study is not the most critical deficit compared to the overall ensemble model spread (Lines 372-374).

In this study, MATSIRO also included the CO₂ fertilization effect, as well as LPJmL (Lines 412-413). Thus, two GWMs take the CO₂ fertilization effect into account, and one GWM

considers DVPs. With regard to LPJmL and MATSIRO, their TFE₅ with three CO₂ scenarios are available: variable CO₂ scenarios for RCP2.6 and RCP8.5 and the fixed CO₂ condition at the 2005 level (the results with the fixed CO₂ condition at the pre-industrial level are not available).

The results of our CO₂ experiment show that the difference between the RCP2.6 run and the fixed CO₂ run is not apparent, indicating the small impact of the CO₂ increase assumed in the low emission scenario (Supplementary Fig. 13a and b). In contrast, the results show differences in several TFE₅ values between the RCP8.5 CO₂ scenario run and the fixed CO₂ run (Supplementary Fig. 13c and d). In terms of RCP8.5, we find earlier TFE₅ in the fixed CO₂ results for several combinations of regions and ensemble members regarding MATSIRO. On the other hand, LPJmL shows both earlier and later TFE₅ due to different CO₂ scenarios (including temperature increase, etc.) because of the mixed impact of the CO₂ fertilization effect and DVPs.

Importantly, the results demonstrate that LPJmL and MATSIRO are not necessarily outliers among GWMs. There is a much larger ensemble spread compared to the impact of the CO₂ fertilization effect and DVPs. Therefore, in response to the reviewer's concern, we would say that the missing CO₂ fertilization effect and DVPs in a couple of GWMs in this study is not the most critical drawback.

Nevertheless, we have acknowledged that the CO₂ fertilization effect and dynamic vegetation processes are crucial areas that the community needs to work further. These processes are clearly written as a source of uncertainty in this TFE estimate and important future research directions (Lines 370-372).

2. (Missing) role of internal/natural climate variability:

The authors' analyses are based on a very limited sample of forcings (four GCMs with a single realization) with no consideration for internal/natural climate variability – which is crucial for analyzing the timing of emergence (e.g., Hawkins & Sutton, 2012; Deser et al., 2012; Hawkins et al. 2014; Lehner et al. 2017). As demonstrated in a comprehensive study by Lehner et al. 2020 (also see earlier study by Hawkins and Sutton, 2019), the role of internal (climate) variability is seemingly important and dominant across several climatic regions, compared to other uncertainty sources (emission scenarios or model structural uncertainty) especially at a short time horizon. Not considering this aspect in the analysis can possibly make an overconfident statement on the timing of emergence (considering the uncertainty around the estimates) and the conclusion drawn in this study that “if society begins reducing greenhouse gas emissions immediately and continuously to follow the stringent mitigation scenario, the initial emergences of such unprecedented conditions would occur before or around the beginning of climate stabilization (the 2040-2050s), calling for urgent adaptation measures.”

Thank you for this valuable comment. We agree and have also cognized the importance of internal variability in estimating the time of the first emergence of unprecedented regional drought conditions. Considering that many preceding studies that investigated the time of emergence of climate change employed 10-30 GCMs to take into account a wider range of uncertainty resulting from internal variability, we must admit that the number of GCMs available in ISIMIP2b is small, although we believe that state-of-the-art bias-corrected GCM projections are crucial for a robust impact assessment of climate change.

Therefore, in consideration of this comment, we carried out a new uncertainty estimation by the block-wise bootstrap method. In the method, a time series from an ensemble member is decomposed into a trend and anomaly, and a large number of time series that reproduce internal variability are generated by random resampling (see Methods. Line 487-514). We performed the TFE analysis for 2 million resampled time series to estimate uncertainty. Although a trend and the maximum variation size are derived from each GCM of the four, this approach enables us to consider the uncertainty in the time series of internal variability. Note that the climate sensitivities of the four GCMs used in ISIMIP2b are relatively well balanced in the spread in CMIP5 models. Uncertainty stemming from internal variability and the model spread is presented as a cumulative probability in Fig. 2b. Additionally, we estimate the likelihood of TFE₅ occurrence over time and discuss regions with robust TFE. Consequently, conclusions were updated, and the manuscript was largely revised in this regard.

Regarding internal variability, we have taken another special precaution in this revision. This revision extended the historical reference period from 35 years to 145 years (1861-2005), the entire historical period in the ISIMIP2b simulation (Lines 431, 463-466, 473-475). This change expanded the number of samples to determine the maximum value for the TFE detection threshold. Thus, the upper bound of the historical internal variability is estimated from four times more samples than considered in the previous manuscript results.

Furthermore, we have applied several measures to reduce the detrimental impacts of internal variability, which have been used from the previous version. Although we suppose that the reviewer understands these measures, let us list them just in case.

[1] To improve the robustness of TFE estimation, we selected a subcontinental scale, not a smaller scale, such as a grid cell scale or basin scale. It is known that the smaller the scale is, the larger the influence of internal variability. Additionally, preceding studies have indicated that spatially aggregated information with large-scale sampling is recommended to improve statistical robustness when investigating changes in extreme events^{1,2} (Lines 114-117, 454-459).

[2] The regional average FDD for each year was serially estimated with a 5-year sampling window. That is, sample values from all grid cells in a region over five years constitute the value. This window sampling filters out interannual variability while retaining multi-decadal variability (Lines 463-466).

[3] The constraint, “for consecutively for a certain number of years”, aims to eliminate an episodic short exceedance due to a large internal variability that could occur without further

climate change, which is not an interest of this study when defining an unprecedented condition relevant to climate change (Line 475).

3. Novelty and concerns on some of the key arguments put forward in the Introductory paragraphs:

The idea for analyzing the time of emergence of unprecedented events has been around for quite some time and reported in several studies (some of them being also appropriately cited in this study). So on this point I do not find this study providing any methodological advancements. Specifically the authors themselves have pointed out the earlier study by Touma et al. on the timing of the emergence of the first unprecedented future drought globally. The authors then make a point that Touma et al. used local runoff data and that “it is also crucial to consider direct human influence on the terrestrial water cycle (e.g., water withdrawal, irrigation and reservoir operation) that cannot be ignored in the context of drought in the Anthropocene” and that “... they employed future projections from General Circulation Models (GCMs) without bias-correction ...”

On these points they argue the use of 5 GWMs that accounted for direct human impacts: such impacts were considered temporally variable only during the historical period (1979-2005, histsoc) but were set to remain constant at the 2005 level during the future period (2006-2099, 2005soc) other than irrigation water demand that was simulated by the GWMs in response to climate change. So essentially the authors mainly consider the impact of varying climate conditions in the future period (and no changes in direct human influences).

Fair enough - my point here why to make a strong argument for the consideration of human influences in the Introductory paragraph – in comparison to previous study - and then latter reading through the Methods we have to find that such considerations were not accounted especially during the future period – for which this study makes all sort of conclusions (changes in FDD, timing of emergence, etc). The same thing goes for the argument of bias-correction in the sense that there is no systematic analysis conducted by the authors to point out that bias-correction would really change their conclusions.

We accepted this important comment, as the novelty of this study was not fully clear in the previous manuscript. Therefore, we reframed and modified the introductory paragraphs (Lines 44-86. 1st – 3rd paragraph). The new paragraphs do not stress technical aspects, such as bias-correction or the direct human influences on the terrestrial water cycle.

Considering two facts: Touma et al. 2015³ investigated only RCP8.5 and did not detect ToE for runoff drought in any regions; however, the literature^{4,5} indicated a regionally significant intensification of drought even at the 1.5 °C level, the novelty of this study lies in the first detection of the timing of emergence on hydrological drought under both RCP2.6 and RCP8.5. In general, most ToE studies focus on RCP8.5 or other higher emission scenarios. However, it is critical to understand whether RCP2.6, in line with the Paris Agreement, is enough to avert unprecedented drought conditions and, if not, how swiftly we need to prepare for undesirable

conditions. In addition, a remarkable difference from previous ToE studies, including Touma *et al.* 2015, is that we do not explore the permanent exceedance from the envelope of historical variability but investigate shorter exceedances. Although previous studies discuss the permanent exceedance presumably to discuss ToE of the change of “climate”, this approach may overlook the important signal for adaptation. That is, the occurrence of a record-breaking condition (e.g., > 1/145-year event in this study) consecutively for several years is abnormal enough from an adaptation point of view, whereas these signals are not the interest of previous studies.

4. Use of SREX regions:

I have difficulties in understanding the use of SREX regions by the authors - especially when they argued in favor of using the river discharge, instead of local runoff. The SREX regions do not respect the hydrological boundary of a river basin. The differences in analysis of TFE5 are very much evident when aggregating results to either SREX regions or to basin-country scale (comparing results shown in Fig. 2 and Supplementary Fig. 8). For example look at the median TFE5 value in SREX region 13 (from Fig. 2 it is around 2040; whereas in Fig 8 it is highly variable between 2050-2080; if I read it right). Anyways the authors need to provide a convincing argument regarding the selection of the regional domain.

Thank you very much for this comment. The use of SREX regions, instead of the basin-scale, aims to reduce uncertainty due to internal variability (Lines 114-117, 454-459). We put a higher priority on internal variability than the smaller scale hydrological boundary. It is widely accepted that the smaller scale an analysis is, the more affected by uncertainty due to internal variability. Fisher *et al.*^{1,2} have argued that a spatially aggregated probability assessment with large-scale sampling enables a more robust investigation of extreme events. Therefore, we applied our analysis at this subcontinental scale because special care is needed regarding uncertainty due to internal variability.

Although basin-scale results were partly included in the previous manuscript, we removed the basin-country scale results in this revision for consistency with other results. The basins we used included 608 basin-country scale units, but some basins were rather small.

We use AR6 regional definition⁶ instead of SREX regions in the revised manuscript because the former is better in the climatic division.

5. Time-periods:

Their need to be a consistency in the time-period used in the study: At Line 80 it states – “... we investigate hydrological drought globally from 1979 to 2099 under RCP2.6 and RCP8.5”; whereas in the caption of Fig. 1 it states, “... compared to a historical period (1971-2005)”.

Thank you for pointing out the inconsistent description of the period. The period described in the figure caption was correct. In this revision, we changed the period of drought analysis from 1861 to 2099 and modified the sentence in the main text properly.

6. Model validations:

Lines 287-88: I could not find in the three cited references 48-50 (note that one of them is published in a conference proceeding), the model validation exercise focusing on FDD – which is subject of this study.

Thank you for carefully checking the citations. The reference about the conference proceeding was a mistake. The incorrect reference was replaced with a published study from *Environmental Research Letters*.

As the reviewer indicated, these citations were not directly about validation on FDD but the validation of simulated river discharge. No validation studies have been published from ISIMIP2b because ISIMIP2b GHM simulations are driven by (bias-corrected) GCM projections. On the other hand, the citations are validation studies from another ISIMIP round (ISIMIP2a) that made use of a couple of reanalysis-based meteorological datasets. ISIMIP2a enables validation in general, but a validation of FDD using ISIMIP2a is not a scope of this study. However, we consider that the model validation studies cited here on river discharge support the validity of the GHMs used in this study. One includes validation of low flow conditions as well.

7. Regional drought characteristics:

Line 333. It is not clear what the authors mean by “...yearly regional drought PDFs were serially estimated with a 5-year sampling windows that ended in each year”.

We modified the sentence, so it is now “the regional average FDD was serially estimated for each year with 5-year sampling windows that ended at each year, i.e., samples during 1861-1865 constituted the regional average FDD for 1865”. (Line 462-463)

Reference:

1. Fischer, E. M., Beyerle, U. & Knutti, R. Robust spatially aggregated projections of climate extremes. *Nat. Clim. Chang.* **3**, 1033–1038 (2013).
2. Fischer, E. M. & Knutti, R. Detection of spatially aggregated changes in temperature and precipitation extremes. *Geomorphology* 547–554 (2014).
doi:10.1002/2013GL058499.Received

3. Touma, D., Ashfaq, M., Nayak, M. A., Kao, S. & Diffenbaugh, N. S. A multi-model and multi-index evaluation of drought characteristics in the 21st century. *J. Hydrol.* **526**, 196–207 (2015).
4. Liu, W. *et al.* Global drought and severe drought-Affected populations in 1.5 and 2°C warmer worlds. *Earth Syst. Dyn.* **9**, 267–283 (2018).
5. Lehner, F. *et al.* Projected drought risk in 1.5°C and 2°C warmer climates. *Geophys. Res. Lett.* **44**, 7419–7428 (2017).
6. Iturbide, M. *et al.* An update of IPCC climate reference regions for subcontinental analysis of climate model data: Definition and aggregated datasets. *Earth Syst. Sci. Data Discuss.* 1–16 (2020). doi:10.5194/essd-2019-258

Reviewer #3 (Remarks to the Author):

A. Main comments

This manuscript present estimates of the Time of Emergence of unprecedented hydrological droughts for regional areas at the global scale. Although the motivation and even some results in some extent are valuable, the methodology chosen for the present study have 3 major issues that would in my opinion prevent its publication in Nature Communications. These 3 points are summarized below and further detailed in specific comments.

We appreciate the time and effort the reviewer has dedicated to reaching a deliberate decision and to providing insightful feedback about our study. Thank you for providing us with another opportunity to improve our study. We revisited our study considering the reviewer comments and have made substantial revisions in response to the reviewer's feedback. Major changes we made include:

1. The main results from the annual scale assessment were replaced with results for the **low-flow season**.
2. Sensitivity analyses of **the scenario of anthropogenic hydrological activities** were carried out.
3. **The introductory paragraphs** were largely revised to describe important assumptions of this study and highlight the fundamental scopes of this assessment.
4. While the uncertainty in TFE had been evaluated only by ensemble model spread in the previous version, we now apply **a bootstrap method to consider uncertainty stemming from internal variability** as well as model uncertainty (see Methods). The results are presented for only regions where ensemble median TFE₅ occurring during the 21st century is statistically robust with $\pm 5\%$ level by the bootstrap test. Our discussion is centered on more robust TFE₅ occurring during the 21st century, with equal to or more than 95% of bootstrap members show TFE₅ during the 21st century (see Methods). Because of this improvement, the main message of this study was also updated.
5. We **extend the historical reference period** from 35 years to 145 years (1861-2005) to better take into account a wider range of uncertainty due to internal variability.

Our point-by-point answers are listed below.

A.1. Definition of hydrological drought

The authors argue that their study advance the state of the art of hydrological drought projection at the global scale as they base their analysis on streamflow and not runoff as done in the literature. I cannot agree more on this point with authors. Unfortunately, they firstly chose to define hydrological drought by anomalies with respect to the annual average cycle, allowing for example to consider lower-than-average high flows in the high-flow season to be considered a drought period. I believe that the motivation of using streamflow comes from a will to better approach real-life behavior and issues, but this choice actually draw things away from actual drought and low-flow management.

We appreciate this comment that gets us to reconsider a seasonal hydrological drought assessment. We agree that an assessment focusing on the low-flow season should be of interest to wide readers of Nature Communications. In response to this opinion, we replaced the main results from the annual scale assessment with an analysis of the low-flow season (Lines 111-112). We investigated 91 days (\approx three months) when river discharge is the least in a year climatologically during the historical period (Supplementary Fig. 1).

On the other hand, because we think that water deficit during the high-flow season could potentially have harmful effects on society and the ecosystem, we also show the results of the annual scale analysis and results for the high-flow season in Supplementary Fig. 3, 4, 11 and 12. Storing water during the high-flow season to prepare for a following dry season is essential for water resource management in many parts of the world. Because our interest lies in the time when climate change would collapse the present hydrological stationary condition and force people to alter the annual cycle of their water use practices historically established, we think the results for annual and the high-flow season are valuable.

Moreover, the set-up used in ISIMIP2b has everything related to water management fixed at the level of year 2005, which make things again away from plausible futures. I have strong reserves on selling such a study with underlying assumptions that could be interpreted in the wrong way. Indeed, the assumption of no change in irrigation area, reservoir operation, and so on (L293-296) prevent one to have a full view of what may happen in terms of hydrological drought as simulated by GHMs that can simulated (and that's a real advance in my view) the anthropogenic water cycle (with abstractions, reservoir operations, and land-use change).

Thank you for this important comment. We avoided additional uncertainty besides that from climate projections to focus on the impact of climate change. High uncertainty remains in those scenarios, and there are no established datasets/approaches to do so. In particular, an obstacle we had is that water management scenarios corresponding to RCP8.5 are not available. To let readers know this important assumption, we added a description about the use of 2005soc for the future period in the introductory paragraph (Lines 106-107).

Because of uncertainty in future water management and data availability, our results represent the state-of-the-art of this field. We believe that our finding that unprecedented regional drought conditions would emerge in specific regions within decades under climate change should be valuable. As we need to improve stepwise, further analyses that include the projection of human activities on the terrestrial water cycle are our future work. On the other hand, in principle, the expansion of irrigation areas to meet future food demand and an increase in water consumption due to socio-economic growth aggravate discharge drought so that TFE will be earlier. Using a water management scenario for RCP2.6 (rcp26soc), which is only available in ISIMIP2b, we demonstrate that the results with rcp26soc show earlier TFE₅ or new detection of TFE₅ for some combinations of regions and ensemble members (Supplementary Fig. 14). Therefore, it is now written that TFE₅, in reality, could be earlier than TFE₅ in 2005soc presented in the main text (Lines 368-369).

This whole issue may be solved in (1) specifying right from the abstract the underlying assumptions in hydrological projections, and (2) specifying right from the abstract what kind of hydrological drought is studied here, or rather looking at low-flows than streamflow anomalies.

Thank you for this suggestion. They are now included in the third sentence in the abstract (Lines 32 and 33). Additionally, we modified an introductory paragraph to describe the underlying assumptions and the definition of hydrological drought. Also, please note that we present the results for the low-flow season in the main section and added results of comparison with rcp26soc.

A.2. Historical Natural variability and TFE

The choice made here by authors to define “unprecedented” is based on exceeding the record value simulated over a historical period, following the choice made by Mora et al. (2013). But the difference between the two studies is that Mora et al. considered the record value over a 145-year period (1860-2005), when the present study considers a record value over a 34-year period (1979-2013). One criticism expressed by Hawkins et al. (2014) on the study by Mora et al. is that they may find a too early emergence of the signal due to their methodological choices. This is evident that the present study would largely fall under the same criticism, as hydrological drought variability cannot be estimated from a so short time period. Depending on the interannual and multidecadal variability expressed in each of the 4 selected GCMs here, exceeding the record value of the 1979-2013, and even for several consecutive years, may statistically happen as early as the 2010s even in a fully stationary climate. This is already a strong point, but my experience in bias-correction showed that the interannual variability from a corrected simulation often increases outside of the reference period used for calibrating the bias-correction. This would add to the issue described above. As a conclusion to this point, and

for an individual GCM projection, the choice made for defining “unprecedented” is highly biased towards and early emergence of the signal.

This issue may be solved in considering a longer reference period (if available in simulations) or considering another threshold for considering expulsion from historical variability.

We appreciate this valuable comment on internal variability, especially about the length of the historical reference period to determine the size of the envelope of historical variability. In response to this comment, we extended the historical reference period from 34 years to 145 years (1861-2005) (Lines 431, 463-466, 473-475). This prolonged reference period is the full length of the historical period of ISIMIP2b and nearly the same length as the reference period of Mora et al. (2013).

A.3. Future Internal variability and TFE

The second main comment is relative to the estimation of future variability and the associated uncertainty, which is quite related to the robustness of TFE values. It is maybe not a coincidence that not that many studies looked at the Time of Emergence of unprecedented droughts at the global scale based on offline runoff/streamflow simulations (as pointed out by the authors), because available ensembles do not have the required size (actually the required size of driving GCM runs) that would allow a robust assessment of such features that are highly dependent on the representation of internal variability. Authors recognize in the discussion (in half a sentence L251) that a larger set of GCMs would be beneficial to the robustness of their findings. The number of driving GCMs is here 4, to be compared to 15, used in a study the authors refer to frequently in the manuscript (Touma et al., 2015). Results from Touma et al. are therefore undoubtedly more robust (statistically speaking only, as there are also other methodological choices that come into play) than the ones presented here that rely only on the temporal pattern of only 4 time series. Works by Deser et al. (2012a,b) clearly showed many years ago the importance of considering large ensembles from a single model (SME) to properly assess future trends. This is all the more important when estimating threshold-dependent features like Times of Emergence. Usually large SME are not available and one take multi-model ensemble (MMEs) instead, hoping that internal variability will be sampled through the use of many single runs. Note that methods currently aim at better describing this GCM internal variability even based on a limited number of runs (see e.g. Evin et al., 2019; Hingray et al., 2019). The TFE of the hydrological signal will be in the present study much too influenced by the actual realization from each of the 4 individual projections to deliver a robust estimate. As consequence, as shown by Fig. 2, the median TFE cannot represent the main mode of the distribution and sometimes actually falls between two modes corresponding to very distant times.

This issue is difficult to solve given the state of global offline simulations that are restricted to a limited number of driving GCM runs. One way forward may lie in looking at innovative recent techniques for estimating internal variability.

We considered this important comment carefully. In response to this valuable comment, we evaluated the uncertainty due to internal variability using the block-wise bootstrap method (see Method. Line 487-514). A time series from an ensemble member was decomposed into a trend and anomaly, and random resampling generated a large number of time series. Then, we carried out a TFE analysis for 2 million resampled time series to estimate uncertainty. Although a trend and the maximum variation size were derived from each of the four GCMs, this approach enabled us to account for the uncertainty in the time series of internal variability. Note that the climate sensitivities of the four GCMs used in ISIMIP2b are relatively well balanced in the spread in CMIP5 models. By this new approach, we evaluated uncertainty in TFE and discussed results focusing on regions with robust TFE. In addition, TFE₅ is discussed from the perspective of likelihood by the bootstrap results. We concluded that a couple of regions show earlier TFE₅ before or around the beginning of climate stabilization of RCP2.6 with approximately 50% or even higher likelihood.

B. Specific comments

L49-51: The literature is indeed scarce in terms of time of emergence for drought-related variables. However, it would be worth mentioning that Vidal et al. (2016) explored the timing of emergence of low-flow conditions outside the natural (internal) hydroclimate variability at the catchment scale for both annual and 30-year averages of low-flow statistics. Pohl et al. (2020) also recently proposed a methodology to better estimate the time of emergence for streamflow variables, that would be worth exploring and at least mentioning.

Thank you very much for recommending these brilliant works. Vidal et al. 2016 is referred to at Line 63. Pohl et al. 2020 is referred at Line 57 and Line 471.

L62-63: Bias-correction has limitations and drawbacks, and may lead to inconsistent changes when applied to future projections, as recently demonstrated by Chen et al. (2020). This should definitely be acknowledged in the manuscript.

Thank you very much for this important information about the limitation of bias-correction for the future period due to the model's climate sensitivity and internal variability. We added a caveat in Method. (Lines 417-418)

L79-83: It is of high importance for the readers to understand that even if the authors look at future streamflow time series, these projections rely on the highly questionable (in terms of correspondence to plausible futures) hypothesis that reservoir location, reservoir operations, and irrigation area will remain at their 2005 level throughout all the 21st century! This part of

the ISIMPI2b protocol is specified in the Methods section, but it would deserve a front row notification to the readers.

We agree with the reviewer's point. An additional sentence was made in the introductory paragraph to stress that we applied 2005soc to purely discuss the impact of climate change (Lines 106-107. 4th paragraph).

In addition, as we mentioned above, a more discussion about this assumption was added to the discussion based on an additional analysis with socioeconomic change under the SSP2 scenario with RCP2.6 (Lines 359-374).

L83-85: Moreover, it is also of high value for the reader to understand here that what is considered as a "hydrological drought" here is a deviation from the average annual regime. That means that lower high flows during the high-flow season will be considered as a drought period and therefore contribute (of course depending on the parameters defined in the Methods section) to the frequency of drought days. This is due to the approach chosen here, based on a daily variable threshold which identifies daily anomalies irrespective of the magnitude of streamflow, as shown recently by Stahl et al. (2020). Of course, this is something that does not reflect what is actually relevant in terms of drought management in the real life. This comment and the preceding one deeply influence the lecture of the results from this study and this should be strongly acknowledged, starting up from the abstract.

Thank you for this suggestion. The main results are now about low flow during the low-flow season, and it is now described in the introductory paragraph (Lines 111-112). We also modified the abstract (Line 33). Additionally, the following sentences have been added to the introductory paragraph describing the methods to briefly explain the definition of hydrological drought used in this study (Lines 107-108).

"Hydrological drought is defined as the condition when daily river discharge is lower than or equal to a daily threshold (see Methods). The threshold includes seasonality."

L93-106, Fig. 1: The 2D color scale is obviously perceptually misleading (and not appropriate for colorblind readers), and this issue has been voiced by many studies now (Staufer et al., 2015), and even in Nature (Hawkins, 2015). Kaye et al. (2012) even provided appropriate 2D colour scales for climate variables. Please reconsider the colour scales of Fig. 1, but also Extended Fig. 2, Supplementary Fig. 2, Supplementary Fig. 3, Supplementary Fig. 6, Supplementary Fig. 7, and in a lesser extent Supplementary Fig. 8.

Thank you for this comment on the inappropriate color scheme. We decided not to use the 2D-colorbar in the revised manuscript. Relative change and ensemble member agreement are

separately presented, using a color scheme proposed in Kaya *et al.* In addition, we do not present Supplementary Figs. 6 and 7.

L108-121: The ISIMIP fast-track experiments allowed a few years ago studies on the multimodel evolution of hydrological droughts and these should be acknowledged in this manuscript (Prudhomme *et al.*, 2014, Giuntoli *et al.*, 2015). Indeed, these studies also provide for example maps of the signal-to-noise ratio of the evolution of runoff drought (with a daily variable and fixed threshold respectively) that should be compared with the ones provided by the authors, notably in terms of hot spots.

The overall similarity in the spatial distribution of changes and signal-to-noise ratio (S2N) are mentioned in the current manuscript (Lines 153-154). We refer to these two previous studies in the Methods section as well (Line 541).

Overall, the spatial distributions of S2N are very similar among the three studies. The studies show higher S2N in hot-spot regions with increased FDD, while areas with those high S2N tended to extend relatively broader in their results. Our result can be considered to have similar validity to Prudhomme *et al.* 2014 and Giuntoli *et al.* 2015.

L128-130: Actually, this precise statistical analysis may lead to similar results even in a fully stationary climate (even if the global homogeneity in significance would be highly improbable). Please consider presenting results from an alternative test set-up or justify the present one better.

This is a very valuable comment. We removed this sentence about the statistical test for the difference between historical period and future 30 years because we found the test is not reasonable to demonstrate the statistically robust impact of climate change under the two RCPs. The reviewer is right. We carried out the same test on a time series under a fully stationary climate, and the difference was statistically significant in a fully stationary climate as well.

L148-149, “The ensemble spread...” I guess there could be an overestimation or underestimation of the probability value with KDE given the censored time interval (2010-2100). Please provide some assessment of this limitation. Additionally, I don’t understand why the left time boundary is 2010 and not 2006 (start of the RCPs) or 2014 (outside of the bias-correction and reference period). Please provide some justification or modify it in the manuscript.

Please be aware that we updated this uncertainty estimation in Figure 2. Instead of the KDE from the limited number of ensemble members, we quantify the uncertainty of internal

variability by 2 million samples from a bootstrap approach. (Please find the details in the next answer and the manuscript).

L142-153, Fig2: This is where the lack of consideration of internal variability in future projections actually emerges in the manuscript... The highly multimodal distribution shown in Fig. 2 for the TFE5 is actually driven by the sole 4 individual time series of the four driving GCMs of the ensemble considered here (4 GCMS x 5 GHMs/LSMs). This means that the internal variability is represented by only 4 series (for a given RCP) whereas it is well known for almost a decade that understanding the internal variability requires much more realizations (GCM runs) to grasp 21st century evolution and trends (Deser et al., 2012a,b). This is all the more important here that the Time of Emergence is highly dependent on this internal variability through the interannual and multidecadal variability of streamflow. Fig. 2 clearly shows the lack of robustness of TFE5 estimates. Were all 5 GHMs have similar performance in representing hydrological droughts, results would show only 4 distinct values derived from the 4 GCM forcing time series (for a given RCP). And using another run from each of the 4 GCMs considered would have led to quite different estimates, which is undoubtedly detrimental to the validity of results presented here. This is discussed in much more detail in the specific comments below.

L161-164: Authors recognize here the large inter-ensemble spread in TFE estimate. Considering both RCPs during the first half of the 21st century indeed increase the robustness of TFE estimates that fall before 2050s. But Fig. 2 actually considers the two RCPs separately (rightly so).

L166-169: To be clear, the two above comments do not imply that I disagree with what is stated here. My point is that the statistical analysis as presented here is not robust enough to fully support it. For example, considering the interquartile range from the ensemble comes down roughly to consider 2 GCMs out of 4...

L170-175: Results commented in this paragraph are the ones that are the less robust here given all comments above.

Thank you very much for these line-by-line comments relating to comments A2 and A3. As we understand that these four comments concern the same issue, let us respond together. Given these comments on internal variability and the small number of realizations, we performed an uncertainty estimate based on a large number of resampled realizations using the block-wise bootstrap method (see Method. Line 487-514). For each original ensemble member, we reproduced 100,000 time series from an original time series and explored TFE₅ from 2 million time series in total. This approach statistically enables an uncertainty estimate considering temporal variation due to internal variability, although a trend and variation size are derived from the four original GCM projections. Note that the climate sensitivities of the four GCMs used in ISIMIP2b are relatively well balanced in the spread in CMIP5 models. Therefore, in this revision, the results of uncertainty evaluation were drastically replaced. We

provide the uncertainty range for all results using the large resampled results, and our discussion focuses on robust TFE₅.

L176-184, Fig. 3: The error bars displayed in the left hand side clearly show the lack of robustness of the signal. This (non-)robustness could be assessed by comparing distributions under both RCPs by e.g. the Mann-kendall two-sided test, and would indeed rarely show any statistical difference across regions (or only by chance given the differences in interannual and multidecadal variability between RCP2.6 and RCP8.5 from any given GCM). Additionally, the light gray error bars are not visible enough on this graph, minimizing their visual impact...

We appreciate this useful suggestion. We carried out a statistical test using the large resampled ensemble data to investigate the difference between RCP2.6 and RCP8.5 (see Method. Line 515-523). The results show that their difference is statistically significant in terms of the median at 18 out of 19 regions where robust median TFE₅ is detected. Please find the details of the statistical tests in the Methods.

The error bars are more visible in the new figure.

L211-224: Given all my above comments on the lack of robustness of estimates of both the historical variability and future internal variability, I am quite reserved on these conclusions on the correspondence of TFEs with DeltaGMT. Supplementary Fig. 4 clearly shows this lack of robustness on temperature and TFE₅.

Similar to the results in Fig. 2, global mean temperature rise (Δ GMT) results were also updated by the result based on the block-wise bootstrap method. Fig. 4b and Extended Data Fig. 4 show the likelihood of TFE₅ as a function of Δ GMT. In addition, importantly, the discussion regarding Δ GMT is drastically updated in the revised version (Lines 305-319).

L222-224: “Extraordinary” is a term used here as a synonym to “not experienced during the reference period”. And this reference period is only 34-year long, which is quite short for understanding hydrological drought variability as a function of natural climate variability (see e.g. Caillouet et al., 2017). So I guess “extraordinary” is not the appropriate term here. As a consequence, as discussed in another comment below, finding unprecedented conditions could perfectly happen even in a complete stationary climate. There are many works on the probability of breaking historical records in the literature, and I would recommend getting deeper in these, all the more given the lack of robustness in this study of estimates of both historical natural variability and future natural (internal) variability.

Thank you very much for this important comment. In terms of “extraordinary” and the short reference period in the previous version, we replace the reference period with a longer period

(1861-2005) in this revision, which is almost the same as the longest reference period (1860-2005) in the previous ToE studies^{1,2}. The current threshold is a 1/145-year extreme drought condition. Thus, the more severe conditions than this threshold consecutively for years can be called unprecedented.

L310-321: I wonder why such pooling technique has been selected here as the Sequent Peak Algorithm (Tallaksen et al., 1997) is much less parameterized and more easily understandable.

Thank you very much for this useful suggestion. The mutual dependence of drought events is an area I would like to explore further. Our approach is the inter-event time approach, according to Tallaksen et al. 1997³. We applied this method because the end of a drought event is intuitively straightforward, and the length of the pooling period is clear. As for the pooling effect, we initially wanted to properly deal with persistent, very small fluctuations of daily river discharge around a threshold. For this purpose, the parameter values in this study are sufficient.

An important difference between IC and SPA is how the end of a drought event is defined. While the end of an event defined by IC is simply the time at which river discharge becomes larger than a certain threshold, SPA defines the end of an event as the time at which the accumulated deficit volume of the event is fully replenished sometime after river discharge becomes larger than a threshold, considering a recovery period. Therefore, it is widely known that SPA tends to detect more multiyear drought when the deficit volume is large, which implies that the pooling period can be far longer than we defined in this study. The recovery process should be important for storage terms, such as reservoirs, but a wider interpretation (water withdrawal from rivers, water quality, tourism, etc.) is also in our minds. Hence, we did not include the recovery processes (of storage term) in this study.

However, we agree that a less parameterized approach should be preferable. We would like to consider an SPA-like approach in our future drought study.

L334: Considering Q80 as a threshold does not correspond to identifying an “extreme phenomenon” as stated in the manuscript, as 73 days per year in average should be identified as dry days (less so with tau and len parameters, I agree, but still).

Thank you for looking into the method in detail. However, I am afraid that we do not agree to this interpretation. “73 days per year on average” is the case one estimates a fixed threshold as the lower 20th percentile river discharge from all daily discharge data through years. However, we applied the daily variable threshold method with 31 days window sampling and 145 years of the reference period. For a certain day of the year, the lower 20th percentile river discharge from 4495 samples is referenced as a threshold. Because of daily variability in one month, this threshold is much stricter than simply 29 times over 145 years. Considering the

daily variability of river discharge and the 145-year long reference period, we think that the Q_{80} used in this study should be strict enough.

C. Technical corrections

L98: Fig. 1, caption: “significant level” → “confidence level”
Accordingly, “significant” was replaced with “confidence”.

L113: “profound increase” ?
“profound” was modified into “pronounced”.

L146, “shading” → shaded
This was modified.

L331: “grids” → “grid cells”
Changed.

L352: “periods” → “period”
“a consecutive periods” was replaced with “a first consecutive period”.

Reference:

1. Mora, C. *et al.* The projected timing of climate departure from recent variability. *Nature* 502, 183–187 (2013).
2. Dirmeyer, P. A. *et al.* Projections of the shifting envelope of Water cycle variability. *Clim. Change* 136, 587–600 (2016).
3. Tallaksen, L. M., Madsen, H. & Clausen, B. On the definition and modelling of streamflow drought duration and deficit volume. *Hydrol. Sci. J.* 42, 15–33 (1997).

Reviewers' Comments:

Reviewer #1:

Remarks to the Author:

I commend the authors on a very successful and compelling revision of the paper. The authors were able to address my comments fully.

The paper provides a clear and comprehensive outlook on global hydrologic drought under anthropogenic change that has not been addressed in such a way before. They show that many vulnerable regions will undergo drought events that can be devastating to communities across the world by the end of the 21st century, and in some cases as early as 2020. They also show that, if the COP Paris Climate Agreement is followed, many regions can be spared these devastating impacts. I believe that this paper will be of high interest to Nature Communications readership, as well as the broader community.

I have no additional comments or changes and I believe that the paper can be published in present form.

Reviewer #2:

Remarks to the Author:

I appreciate the authors efforts in revising their manuscript which have very much improved in terms of both the significance of the work and the clarity of the presentation. I have only few comments and clarifications to this revision round.

The first one is related with the use of AR6 regions: I continue to question the authors intention of using the (43) AR6 regions to aggregate and summarize the FDD results - since these regions are not motivated by any hydrological reasoning. In response to my previous comment on this topic (and also the arguments given in the text) the authors build their argument and cited and referred two papers of Fisher et al. (43 and 89 in main text) – please note that both of these Fisher et al works are motivated for climate variables (precipitation and temperature extremes) unlike the authors work which focus on streamflow (or derived drought parameters) and that at any point on a river network, streamflow is already an aggregated behavior considering all their upstream contributions. I could understand if the authors could have considered instead of the streamflow (FDDs), a grid specific runoff based FDDs calculated from just a given grid generated runoff contribution – that can in analogous to precipitation or temperature can be aggregated over a domain.

Also, there is a clear issue in averaging the grid-specific streamflow (FDD) over a given AR6 region using the latitudinal grid area difference as a weighing factor. Again, the same argument holds – this procedure is good and reasonable for other climate variables (precipitation or temperature) but not for the streamflow derived parameters. Certainly, the weighing factor, taken as grid-specific latitudinal information, is not correct approach – it ignores the very fundamental fact that streamflow at a particular grid is not only its own contribution but also all their upstream locations (grids) draining to this particular grid. Accordingly, I still find it problematic the use of AR6 regions (or even earlier SEREX ones) and the weighting approach in the authors work.

The second one is related with quantifying the internal variability component: I understand the choice of the block-wise bootstrap method to quantify the uncertainty due to internal variability. There are clearly some choices that need to be made and these are not clear to me from the current text.

1. What kind of block bootstrapping method was used? Simple, non-overlapping, moving block?
2. Why was the quadratic function used for estimating trend? Did you try other methods?
3. Did you check that the residuals have any other remaining (consistent) trends?

4. Choice of block size – is motivated by autocorrelation value of 0.3. This is still a very high value considering that the regional average FDD was serially estimated for each year with 5-year sampling windows. This is crucial information – the block size – for this kind of bootstrapping; and thereby the analysis of internal variability. Better motivate the choice of block size. Consider showing a plot where this condition can/can't meet to qualify “the 5-year-lag autocorrelations are lower than 0.3 for most of the time series of all ensemble members and regions.”

Minor remarks

Line 34: I would suggest to spell out explicitly the names of those four regions here in the Abstract.

Line 460: “whereas grid cells with glacial ice.” – the sentence is not complete.

Reviewer #3:

Remarks to the Author:

This manuscript is a revised version of a submission I previously reviewed. While recognizing the marked relevance of the topic addressed, I had raised three important issues that prevented this initial submission to be published in Nature Communications. I suggested methodological changes to address these issues, but I was not sure of their feasibility for this particular study. I was therefore much favorably impressed by the way authors tackled these issues one by one and produced a revised manuscript that is in my view much more scientifically sound and that conveys much more robust messages. The authors should therefore be congratulated for their hard work on addressing the issues I raised. In short : (1) the manuscripts better define what kind of hydrological drought is considered here, by looking at abnormally low flows during the low-flow period only, (2) it now takes into account a 145-year long baseline (instead of 31-year previously) for defining the most severe historical drought, and (3) it now actually deals with natural variability in projections with a relevant bootstrapping method that makes the most of the few existing climate trajectories. Most of the specific comments have also been addressed in a satisfactory way.

I have only one comment remaining on the definition of the daily variable threshold. The chosen approach indeed mixes for a given Julian day the year-to-year variability and the day-to-day variability (within a month centered around the Julian day considered). This is definitely not standard practice in the many studies using a daily variable threshold where the year-to-year variability is taken into account (at either the daily or monthly scales, by the way), and a smoothing is performed afterwards (see Van Loon, 2015; Caillouet et al., 2017). The standard method therefore directly identifies the threshold with a constant probability of exceedance for any Julian day over the period considered for calculating the threshold. As noted by the authors in the reply to my comments, this is not the case following the methodological choices made in the manuscript. These choices therefore prevent one assessing this probability of exceedance for the method's parameters, as it depends on the day-to-day variability of the considered initial time series. It is therefore impossible for the reader to have a clear idea of the level of extremeness of the events that cross this threshold. As adopting the standard way of calculating this threshold would imply redoing all the analyses, I would simply recommend assessing a posteriori the average (over the cells and days of the year) of the probability of exceedance over the reference period (1861-2005) that actually derives from the chosen calculations with a Q80 over a 31-day moving window. The question is basically : is it more like 85 or 95? Brief insights into the variability (over the cells and days of the year, and especially for the low-flow period of course) of this probability would also be welcome, naturally. Such a calculation would be easily done based

As a conclusion, I would like to thank again the authors, and I am glad that my comments led to these substantial changes to the initial manuscript, changes that led in my view to much more robust findings and messages. I would therefore be happy to recommend publication of the revised manuscript in Nature Communications, provided that the authors address this final small comment.

References

Caillouet, L., Vidal, J.-P., Sauquet, E., Devers, A. & Graff, B. (2017) Ensemble reconstruction of spatio-temporal extreme low-flow events in France since 1871. *Hydrology and Earth System Sciences*, 21 (6), 2923-2951. <https://doi.org/10.5194/hess-21-2923-2017>

Van Loon, A. F. (2015) Hydrological drought explained. *WIREs Water*, 2 (4) 359-392, <https://doi.org/10.1002/wat2.1085>

Response Letter

Reviewer #1 (Remarks to the Author):

I commend the authors on a very successful and compelling revision of the paper. The authors were able to address my comments fully.

The paper provides a clear and comprehensive outlook on global hydrologic drought under anthropogenic change that has not been addressed in such a way before. They show that many vulnerable regions will undergo drought events that can be devastating to communities across the world by the end of the 21st century, and in some cases as early as 2020. They also show that, if the COP Paris Climate Agreement is followed, many regions can be spared these devastating impacts. I believe that this paper will be of high interest to Nature Communications readership, as well as the broader community.

I have no additional comments or changes and I believe that the paper can be published in present form.

Response: We thank the reviewer for going through our revised manuscript. We are very pleased that the reviewer found our revisions to be reasonable and adequate and found our study to be valuable for the readership of Nature Communication.

Reviewer #2 (Remarks to the Author):

I appreciate the authors efforts in revising their manuscript which have very much improved in terms of both the significance of the work and the clarity of the presentation. I have only few comments and clarifications to this revision round.

Response: We thank the reviewer for reading the revised version of our manuscript and providing comments helping us to further strengthen our paper. We have further revised the manuscript following the suggestions from the reviewer. A point-by-point response to each of the comments is provided in the following text.

The first one is related with the use of AR6 regions: *I continue to question the authors intention of using the (43) AR6 regions to aggregate and summarize the FDD results - since these regions are not motivated by any hydrological reasoning. In response to my previous comment on this topic (and also the arguments given in the text) the authors build their argument and cited and referred two papers of Fisher et al. (43 and 89 in main text) - please note that both of these Fisher et al works*

are motivated for climate variables (precipitation and temperature extremes) unlike the authors work which focus on streamflow (or derived drought parameters) and that at any point on a river network, streamflow is already an aggregated behavior considering all their upstream contributions. I could understand if the authors could have considered instead of the streamflow (FDDs), a grid specific runoff based FDDs calculated from just a given grid generated runoff contribution - that can in analogous to precipitation or temperature can be aggregated over a domain. Also, there is a clear issue in averaging the grid-specific streamflow (FDD) over a given AR6 region using the latitudinal grid area difference as a weighing factor. Again, the same argument holds - this procedure is good and reasonable for other climate variables (precipitation or temperature) but not for the streamflow derived parameters. Certainly, the weighing factor, taken as grid-specific latitudinal information, is not correct approach - it ignores the very fundamental fact that streamflow at a particular grid is not only its own contribution but also all their upstream locations (grids) draining to this particular grid. Accordingly, I still find it problematic the use of AR6 regions (or even earlier SEREX ones) and the weighting approach in the authors work.

Response: We thank the reviewer for providing us with the opportunity to further reconsider this important issue. In response to the reviewer's comment, we replaced the use of AR6 regions with a more hydrologically reasonable regional definition derived from HydroBASINS¹ level 2 data (Supplementary Fig. 2) and thoroughly updated our results. By globally delineating river basins based on topography at multiple scales, HydroBASINS provides regional tiles defined by river basin shapes (Lines 123-124, 486-488). The dataset first divides the global land area into nine continents (level 1), further splits each continent into up to 9 large subunits (level 2), and hierarchically nests subbasins at different scales ranging from tens to millions of square kilometers. Notably, as we discussed in the previous round of revisions, large-scale sampling is crucial, so we utilized the level-2 HydroBASINS product to consider a total of 59 regions. Nonetheless, importantly, the findings and messages derived in this study are basically unchanged even after the regional definition is changed. Our understanding is that the reviewer was concerned that changes in the regional drought characteristics of a certain AR6 region may be influenced by hydrological changes in upstream areas in other AR6 regions. Thus, we believe that the HydroBASINS product provides a more straightforward analysis that allows us to better understand regional impacts and their causes.

In terms of the weighted averages, first, we would like to stress that we do not average the grid-specific streamflow itself but instead average the grid-specific frequency of drought days (FDDs). Second and most importantly, we kindly ask for the reviewer's understanding that our focus regarding water resources lies in the consequential climate hazards that are projected to occur and that human society will have to cope with at each grid cell, regardless of where these impacts

originate from. With this as our focus, we use river discharge instead of local runoff; as the reviewer pointed out, grid-scale water resources can be influenced by changes in upstream areas. Additionally, to assess hydrological drought based on river discharge measurements, one might lump a basin and use the river discharge data only at the river outlet to obtain regional representative information; however, this method omits the regional heterogeneity in drought exposure that can exist within a region. The larger a basin is, the more important heterogeneity is. Hence, we conducted the drought detection and the frequency estimation separately at each grid cell based on a grid-specific threshold, adopted FDD as a hazard proxy, and sampled FDD values from all grid cells in a given region to estimate the regional statistics for the TFE analysis. Third, we still assume that the area weights are sufficiently valid to define a regional representative value because of the occasionally large area differences among grid cells in a given region. For instance, in the Southwestern South America region, which is wide in the north–south direction and shows the earliest and most robust TFE₅ in our study, the area of the southernmost grid cell is only 60% that of the northernmost grid cell due to the latitudinal effect. Thus, considered these grid cells to be equal when defining a regional proxy from grid values would lead to errors. One may further apply other ways to define grid weights, such as using population affected or seasonal average river discharge values, depending on the specific study interests, but we selected a rather simple approach to consider only area differences. Finally, if this comment suggests a different weighting scheme in which the potential impacts of drought conditions at a grid cell downstream or upstream of a given region are considered, we assume that without such a weighting scheme, the new regional definition and explicit river discharge simulations conducted in the study solve this issue. As a result, we believe that the use of the new regional definition and the area-weighted averaging scheme can be accepted when calculating regional statistics.

The second one is related with quantifying the internal variability component: I understand the choice of the block-wise bootstrap method to quantify the uncertainty due to internal variability. There are clearly some choices that need to be made and these are not clear to me from the current text.

Response: Thank you very much for the comments helping us to clarify the method descriptions; these revisions are described in more detail below.

1. What kind of block bootstrapping method was used? Simple, non-overlapping, moving block?

Response: We simply split a time series of 115 years into 23 nonoverlapping blocks. We added a sentence in the Methods section (Lines 531) to clarify this point.

2. Why was the quadratic function used for estimating trend? Did you try other methods?

Response: We tried various functions from linear to quintic functions and found that the quadratic function offered more versatility than other functions for our 59 regions and 20 GHM and GCM combinations. We added Supplementary Figs. 16 and 17 as sample plots derived for regions with CWatM forced by Hadgem2-ES climate forcing. Higher-order ($>3^{\text{rd}}$) functions than the quadratic function can include natural-variability-like behaviors as a part of the trend, and these functions often show significantly rapid changes at the beginning or/and end of the study period. We sought a more gradual curve when deriving long-term trends to avoid overfitting. On the other hand, a linear function is too simple to depict any unique long-term trend, such as those observed in Southeast North America and the Middle East under RCP2.6, in which peaks can be observed in the middle of the 21st century. To obtain a better-fitting curve, we may have been able to select the best function individually for each case, but it is difficult to prove that the variability in a long-term trend described by higher-order functions is not natural variability. Thus, we defined gradual quadratic long-term trends and residuals consistently for all cases.

3. Did you check that the residuals have any other remaining (consistent) trends?

Response: The following Figures R1 and R2 present the residual time series derived from the quadratic fitting curves. The black lines describe the ensemble median values of the residual terms, and the shading denotes the 5-95% range in the ensemble spread. We selectively presented time series for the 19 regions in which the original time series showed clear trends (Fig. 1b and Extended Fig. 3) and TFE₅ occurrence was detected (Fig. 2a). Because both the ensemble medians and the shaded areas oscillate around zero and no systematic trend is observed throughout the period in any region, we do not believe that any critical trend remains in the residual terms. Thus, we assume the quadratic fitting is sufficiently valid.

Figure R1 | Residual time series derived from the quadratic fitting curves of 19 regions under RCP2.6.

Figure R2 | The same figure as Fig. R1 but under RCP8.5.

4. Choice of block size - is motivated by autocorrelation value of 0.3. This is still a very high value considering that the regional average FDD was serially estimated for each year with 5-year sampling windows. This is crucial information - the block size - for this kind of bootstrapping; and thereby the analysis of internal variability. Better motivate the choice of block size. Consider showing a plot where this condition can/can't meet to qualify "the 5-year-lag autocorrelations are lower than 0.3 for most of the time series of all ensemble members and regions."

Response: Thank you for this important suggestion. We added Supplementary Fig. 18 to present the cumulative probability of the effective decorrelation time (Te) at which the autocorrelations become lower than 0.3. The cumulative probability begins to be saturated when Te is approximately 5 years, and more than 70% of combinations of regions, GHMs, and GCMs reach autocorrelations lower than 0.3 when Te is equal to 5 years under RCP2.6 and RCP8.5.

Minor remarks

Line 34: I would suggest to spell out explicitly the names of those four regions here in the Abstract.

Response: Thank you for this comment. We listed the full names of the three most important regions in the abstract.

Line 460: “whereas grid cells with glacial ice.” - the sentence is not complete.

Response: We replaced “whereas” with “except”.

Reviewer #3 (Remarks to the Author):

This manuscript is a revised version of a submission I previously reviewed. While recognizing the marked relevance of the topic addressed, I had raised three important issues that prevented this initial submission to be published in Nature Communications. I suggested methodological changes to address these issues, but I was not sure of their feasibility for this particular study. I was therefore much favorably impressed by the way authors tackled these issues one by one and produced a revised manuscript that is in my view much more scientifically sound and that conveys much more robust messages. The authors should therefore be congratulated for their hard work on addressing the issues I raised. In short : (1) the manuscripts better define what kind of hydrological drought is considered here, by looking at abnormally low flows during the low-flow period only, (2) it now takes into account a 145-year long baseline (instead of 31-year previously) for defining the most severe historical drought, and (3) it now actually deals with natural variability in projections with a relevant bootstrapping method that makes the most of the few existing climate trajectories. Most of the specific comments have also been addressed in a satisfactory way.

Response: We thank the reviewer for reading our revised manuscript and providing this highly positive evaluation. We are glad that the reviewer believes that our revisions successfully improved the scientific reasonability and robustness of this study.

I have only one comment remaining on the definition of the daily variable threshold. The chosen approach indeed mixes for a given Julian day the year-to-year variability and the day-to-day variability (within a month centered around the Julian day considered). This is definitely not standard practice in the many studies using a daily variable threshold where the year-to-year variability is taken into account (at either the daily or monthly scales, by the way), and a smoothing is performed afterwards (see Van Loon, 2015; Caillouet et al., 2017). The standard method therefore directly identifies the threshold with a constant probability of exceedance for any Julian day over the period considered for calculating the threshold. As noted by the authors in the reply to my comments, this is not the case following the methodological choices made in the manuscript. These

choices therefore prevent one assessing this probability of exceedance for the method's parameters, as it depends on the day-to-day variability of the considered initial time series. It is therefore impossible for the reader to have a clear idea of the level of extremeness of the events that cross this threshold. As adopting the standard way of calculating this threshold would imply redoing all the analyses, I would simply recommend assessing a posteriori the average (over the cells and days of the year) of the probability of exceedance over the reference period (1861-2005) that actually derives from the chosen calculations with a Q80 over a 31-day moving window. The question is basically : is it more like 85 or 95? Brief insights into the variability (over the cells and days of the year, and especially for the low-flow period of course) of this probability would also be welcome, naturally. Such a calculation would be easily done based

Response: We admit that the applied daily variable threshold method¹ is not straightforward in terms of the “general” probability of exceedance; however, theoretically, this probability is Q80 at each Julian day of the year (DOY) according to our definition. Following the reviewer’s suggestion, we performed an additional analysis and the results are presented in Supplementary Fig. 15; we also added one phrase regarding the results to the Methods section (Lines 464-467). We estimated the average probability of exceedance (PoE) of our Q80 values over the reference period without considering day-to-day variabilities. First, we created a time series of 31-day moving-averaged river discharge totals from 1861-2005; then, we estimated the PoE of our Q80 at each DOY compared to the year-to-year variabilities using 145 samples. Because PoE shows regionality, we averaged PoE only over the DOY and presented a global map. Owing to the day-to-day variability, our approach allows for stricter drought detection, as it can refer to lower river discharge as the Q₈₀ threshold compared to the standard approach. The histogram (Supplementary Fig. 15a) shows that our Q80 corresponds to less frequent events in more than 90% of grid cells. The red-colored grid cells in Supplementary Fig. 15b are considered to be highly influenced by day-to-day variabilities. We hope this supporting information addresses the reviewer’s final concern.

As a conclusion, I would like to thank again the authors, and I am glad that my comments led to these substantial changes to the initial manuscript, changes that led in my view to much more robust findings and messages. I would therefore be happy to recommend publication of the revised manuscript in Nature Communications, provided that the authors address this final small comment.

Reviewers' Comments:

Reviewer #2:

Remarks to the Author:

I congratulate the authors for their efforts in revising and considering my previous comments/suggestions. The work has greatly improved both in clarity and content and possibly can go through the next phase of the publication.

I have one remaining remark on the choice of block size – first thanks to the authors for considering my earlier suggestion on providing a plot – in this context I would suggest the authors to show results as the spatial plots rather than the bar plots (as given now in Supplement Figure 18). This is to better see as in which regions does the autocorrelation criteria (of 0.3) does not hold under each RCP; and what would be the implications if some of these regions would overlap with ones that are crucial and highlighted (Figures 2, 4) in the main text.

Response Letter

Reviewer #2 (Remarks to the Author):

I congratulate the authors for their efforts in revising and considering my previous comments/suggestions. The work has greatly improved both in clarity and content and possibly can go through the next phase of the publication.

I have one remaining remark on the choice of block size – first thanks to the authors for considering my earlier suggestion on providing a plot – in this context I would suggest the authors to show results as the spatial plots rather than the bar plots (as given now in Supplement Figure 18). This is to better see as in which regions does the autocorrelation criteria (of 0.3) does not hold under each RCP; and what would be the implications if some of these regions would overlap with ones that are crucial and highlighted (Figures 2, 4) in the main text.

Response:

We thank the reviewer for going through our manuscript again and providing the positive evaluation with our last revision.

In response to this comment, we replaced the cumulative bar graphs with (i) regional boxplots concerning the correlation coefficient at lag=5 and (ii) global maps summarizing regional median values (Supplementary Figs. 23 and 24). A boxplot is composed of 20 ensemble members for each region. Hence, the figures present regional characteristics regarding the validity of our decision concerning the block size, i.e., the five-year nonoverlapping block-wise resampling method. In addition, we added the following brief description discussing the special characteristics and the validity.

“A few ensemble members for specific regions, such as parts of Asia and the Middle East, show higher values than others, but their correlations are also not strong with the log size.” (Lines 492-494)